# TOWARDS COMPLETELY EXPRESSIVE CAPACITY OF MIXED MULTI-AGENT Q VALUE FUNCTION

## ABSTRACT

Value decomposition is an efficient approach to achieving centralized training with decentralized execution in fully cooperative Multi-Agent Reinforcement Learning (MARL) problems. Recently, Strictly Monotonic Mixing Function (SMMF) has gained widespread application in value decomposition methods, but SMMF could suffer from convergence difficulties for the representational limitation. This paper investigates the circumstances under which the representational limitation occurs and presents approaches to overcome it. We begin our investigation with Linear Mixing Function (LMF), a simple case of SMMF. Firstly, we prove that LMF is free from representational limitation only in a rare case of MARL problems. Secondly, we propose a two-stage mixing framework, which includes a difference rescaling stage after SMMF to complete the representational capability. However, the capacity could remain unrealized for the cross interference between the representation of different action-values. Finally, we introduce gradient shaping to address this problem. The experimental results validate the expressiveness of LMF and demonstrate the effectiveness of our proposed methods.

## 1 INTRODUCTION

Centralized training with decentralized execution (CTDE) (Oliehoek et al., 2008; Foerster et al., 2016) shows surprising performance and great scalability in challenging fully cooperative multi-agent tasks (Tan, 1993b). Such tasks only provide team rewards. Each agent is expected to deduce its own contribution to the team, which introduces the problem of credit assignment (Lowe et al., 2017; Foerster et al., 2018). A simple and efficient approach to achieving credit assignment in the CTDE paradigm for value-based Multi-Agent Reinforcement Learning (MARL) is value decomposition. Through value decomposition, a joint Q value function is factorized into a set of local Q functions. The joint Q function is trained centrally to estimate the action-values (i.e., the expectation of accumulated rewards), whereas the local Q value functions are assigned to agents for decentralized execution. The projection from local Q values to joint Q values is defined as the mixing function. A critical principle of value decomposition is Independent Global Max (IGM), i.e., the identity between the joint greedy action and the set of local greedy actions (Son et al., 2019). Strictly Monotonic Mixing Function (SMMF) is simple and exactly satisfies the IGM. As a result, SMMF is widely applied in value decomposition methods (Rashid et al., 2020b;a).

However, the joint Q value function of SMMF is incapable to completely represent the correct action-values in arbitrary MARL tasks, known as the problem of representational limitation (Rashid et al., 2020a). A widely discussed case of SMMF is Linear Mixing Function (LMF). The representational limitation of LMF would introduce sub-optimal fixed-points (Wan et al., 2021) or even leads to the divergence of the joint Q value function trained by Q-learning value iteration (Wang et al., 2020a). The complete representation of the action-values is determined both by the complexity of action-value function and the representational capacity of the joint Q value function. The former is an inherent property of the task and the latter depends mainly on the form of the mixing function. In this paper, we investigate the representational limitation from both perspectives. Our investigations aim to answer the following questions: (1) In what kind of problems the representational limitation occurs for LMF? (2) How to obtain mixing functions with complete representational capacity?

The joint Q value function of LMF is incapable to represent the action-value functions that are not linearly factorizable. Previous works (Son et al., 2019; Rashid et al., 2020a) generally focus on

the solutions to the representational limitation but neglect in what kind of problems the limitation occurs. In section 3, we prove that the action-value function is linearly factorizable *if and only if* the problem can be modelled as a decomposable MMDP, which is a rare case of MARL problems. In words, LMF suffers from the representational limitation in most MARL tasks. In section 4.1, we find the representational limitation of SMMF stems from the bounded difference between the joint Q values of the greedy and other actions. Therefore, we propose a two-stage mixing framework, namely Mixing for Unbounded Difference (MUD), which mixes and rescales the output differences of multi-channel SMMFs into unbounded. MUD achieves complete representational capacity under the IGM constraint. However, such capacity could remain unrealized for the representational cross interference between different action-values, which is a fundamental problem of value decomposition. To address the problem, in section 4.2, we introduce gradient shaping in MUD and propose two novel mixing functions. In experiments, we design a toy game to verify the linear factorizability of the action-value function in decomposable MMDP. Besides, we evaluate the proposed mixing functions across various benchmarks, demonstrating their superiority over baselines.

There are three contributions in this paper: (1) we are the first to prove a sufficient and necessary condition for the occurrence of the representational limitation in LMF; (2) we propose a mixing framework with complete representational capacity under the IGM constraint; (3) we propose a fundamental problem of value decomposition, namely optimal representation interference. We also propose two novel mixing functions, which outperform baselines and address the problem.

## 2 PRELIMINARY

### 2.1 DEC-POMDP

A fully cooperative Multi-Agent Reinforcement Learning (MARL) problem can be modelled by the Decentralized Partially Observable Markov Decision Process (Dec-POMDP), which is usually described by a tuple $G = \langle \mathcal{S}, \mathcal{U}, \mathcal{P}, r, Z, O, n, \gamma \rangle$ (Guestrin et al., 2001; Oliehoek et al., 2016; Seuken & Zilberstein, 2008). $s \in \mathcal{S}$ denotes the global state of the environment. A local observation $z_a \in Z$ is assigned to agent $a \in A \equiv \{1, 2, \cdots, n\}$ according to the observation function $O(s, a) : \mathcal{S} \times A \rightarrow Z$. After receiving $z_a$, each agent chooses an action $u_a \in \mathcal{U}_a$ based on its local policy $\pi_a(u_a|\tau_a) : \mathcal{T} \times \mathcal{U}_a \rightarrow [0, 1]$, where $\tau_a \in \mathcal{T} \equiv (Z \times \mathcal{U}_a)^*$ is the local observation-action history. After the execution of the joint action $\boldsymbol{u} = \{u_1, \cdots, u_n\}$, a team reward $r$ and the next state $s'$ are generated by the reward function $r(s, \boldsymbol{u}) : \mathcal{S} \times \mathcal{U} \rightarrow \mathcal{R}$ and transition function $\mathcal{P}(s'|s, \boldsymbol{u}) : \mathcal{S} \times \mathcal{U} \times \mathcal{S} \rightarrow [0, 1]$, respectively, where $\boldsymbol{u} \in \mathcal{U} \equiv \mathcal{U}_1 \times \cdots \times \mathcal{U}_n$. $\gamma \in [0, 1)$ is a discount factor. We denote the set of actions except $u_a$ by $u_{\setminus a}$, i.e., $u_{\setminus a} \cup u_a = \boldsymbol{u}$. We only consider the fully observable setting for theoretical analysis, which can be modelled by Multi-agent Markov Decision Process (MMDP). An MMDP is usually described by tuple $\mathcal{MG} = \langle \mathcal{S}, \mathcal{U}, \mathcal{P}, r, n, \gamma \rangle$.

In value-based MARL, an approximate function is trained to estimate the action-values by value iteration. We discriminate the *action-value function* and its approximation by $\mathcal{Q}(s, \boldsymbol{u})$ and $Q_{tot}(s, \boldsymbol{\tau}, \boldsymbol{u})$, respectively, where $\boldsymbol{\tau} = \{\tau_1, \cdots, \tau_n\} \in \mathcal{T}^n$. The latter is named *joint Q value function*. The action-value function is defined as the expectation of accumulated rewards, i.e., $\mathcal{Q}(s_t, \boldsymbol{u}_t) := \mathbb{E}_{s_{t+1:\infty}, \boldsymbol{u}_{t+1:\infty}}[R_t|s_t, \boldsymbol{u}_t]$, where $R_t = \sum_{i=0}^{\infty} \gamma^i r_{t+i}$. The action with the maximal action-value is defined as the *optimal action*, which is denoted by $\boldsymbol{u}^* := \arg\max_{\boldsymbol{u}} \mathcal{Q}(s, \boldsymbol{u})$.

### 2.2 VALUE DECOMPOSITION

Let $Q_a(\tau_a, u_a) : \mathcal{T} \times \mathcal{U}_a \rightarrow \mathcal{R}$ denote the local Q function of agent $a$ ($a \in [1, n]$). Through value decomposition, the joint Q function is factorized into a set of local Q value functions as

$$Q_{tot}(s, \boldsymbol{\tau}, \boldsymbol{u}) = \mathcal{F}(\cup_{a=1}^n Q_a, s, \boldsymbol{\tau}, \boldsymbol{u}) \tag{1}$$

where we abbreviate $Q_a(\tau_a, u_a)$ as $Q_a$. $\mathcal{F}$ is named mixing function. The action with the maximal *local Q value* is defined as the *greedy action*, i.e., $u_{a,gre} := \arg\max_{u_a} Q_a(\tau_a, u_a)$. There are two critical concepts of value decomposition: Independent Global Max (IGM) and monotonicity.

**IGM**. Given the joint Q function $Q_{tot}(s, \boldsymbol{\tau}, \boldsymbol{u}) = \mathcal{F}(\cup_{a=1}^n Q_a, s, \boldsymbol{\tau}, \boldsymbol{u})$, if $\forall s \in \mathcal{S}$,

$$\arg\max_{\boldsymbol{u} \in \mathcal{U}} Q_{tot}(s, \boldsymbol{\tau}, \boldsymbol{u}) = \left\{ \arg\max_{u_1 \in \mathcal{U}_1} Q_1(\tau_1, u_1), \cdots, \arg\max_{u_n \in \mathcal{U}_n} Q_n(\tau_n, u_n) \right\} \tag{2}$$

we say $\mathcal{F}$ satisfies the IGM. The IGM ensures decentralized executions (according to the largest local Q values) always bring the best estimated group interests (i.e., the largest joint Q values).

**Monotonicity**. Given the joint Q value function $Q_{tot}(s, \boldsymbol{\tau}, \boldsymbol{u}) = \mathcal{F}\left(\cup_{a=1}^{n} Q_a(\tau_a, u_a), s, \boldsymbol{\tau}, \boldsymbol{u}\right)$, we say the mixing function $\mathcal{F}$ is *monotonic* if $\forall s \in \mathcal{S}, \forall a \in [1, n]$,

$$\frac{\partial Q_{tot}(s, \boldsymbol{\tau}, \boldsymbol{u})}{\partial Q_a(\tau_a, u_a)} \geq 0 \tag{3}$$

Specially, given a monotonic mixing function $\mathcal{F}$ and $\forall u_a, u_a' \in \mathcal{U}_a$, where $Q_a(\tau_a, u_a) \neq Q_a(\tau_a, u_a')$. We say $\mathcal{F}$ is *strictly monotonic* if

$$\frac{Q_{tot}(s, \boldsymbol{\tau}, u, u_{\setminus a}) - Q_{tot}(s, \boldsymbol{\tau}, u', u_{\setminus a})}{Q_a(\tau_a, u) - Q_a(\tau_a, u')} \geq 0 \tag{4}$$

The mixing functions of VDN (Sunehag et al., 2017) and QMIX (Rashid et al., 2020b) are strictly monotonic. Strictly monotonic mixing functions naturally satisfy the IGM (the proof is available in Appendix A). Examples of monotonic and strictly monotonic mixing functions are shown in Fig.1.

Figure 1: Examples of (a) monotonic mixing function $Q_{tot}(s, \boldsymbol{\tau}, \boldsymbol{u}) = \sum_{a=1}^{n} Q_a(\tau_a, u_a) + b(s, \boldsymbol{u})$ and (b) strictly monotonic mixing function $Q_{tot}(s, \boldsymbol{\tau}, \boldsymbol{u}) = \sum_{a=1}^{n} Q_a(\tau_a, u_a)$, where $b(s, \boldsymbol{u})$ is a central bias. Only the latter satisfies the IGM.

## 3 EXPRESSIVE CAPACITY OF LINEAR MIXING FUNCTION

Linear Mixing Function (LMF) is strictly monotonic and satisfies the IGM. The joint Q value function of LMF equals $Q_{tot}(s, \boldsymbol{u}, \boldsymbol{\tau}) = \sum_{a=1}^{n} Q_a(u_a, \tau_a)$. In this section, we investigate in what kind of tasks the representational limitation occurs for LMF. For the convenience of theoretical analyses, we assume that 1) the task is fully observable; 2) the training data follows on-policy distribution.

### 3.1 DECOMPOSABLE MMDP: THE TASK WITH LINEARLY FACTORIZABLE ACTION-VALUES

A fully cooperative, fully observable MARL problem can be modelled by Multi-agent Markov Decision Process (MMDP), which is usually described by a tuple $\mathcal{MG} = <\mathcal{S}, \mathcal{U}, \mathcal{P}, r, n, \gamma>$. Let $\hat{\mathcal{S}}$ and $\hat{\mathcal{U}}$ denote a subset of the state space and a non-empty proper subset of the joint action space, respectively, i.e., $\hat{\mathcal{S}} \subset \mathcal{S}, \hat{\mathcal{U}} \subsetneq \mathcal{U}$ and $\hat{\mathcal{U}} \neq \emptyset$. We introduce the decomposability of an MMDP.

**Definition 3.1.** Given an MMDP $\mathcal{MG} = <\mathcal{S}, \mathcal{U}, \mathcal{P}, r, n, \gamma>$, if there exists $\{\hat{\mathcal{S}}_1 \times \hat{\mathcal{U}}_1, \cdots, \hat{\mathcal{S}}_k \times \hat{\mathcal{U}}_k\}$ ($k \geq 2; \hat{\mathcal{U}}_i \neq \hat{\mathcal{U}}_j$ for $i \neq j$ $(i, j \in [1, k])$), such that $\forall (s_t, \boldsymbol{u}_t) \in \mathcal{S} \times \mathcal{U}$

    1  $\forall i \in [1, k], \mathcal{P}(\hat{s}_{i,t+1}|\hat{s}_{i,t}, \hat{u}_{i,t}) = \mathcal{P}(\hat{s}_{i,t+1}|s_t, \boldsymbol{u}_t)$, where $(\hat{s}_{i,t}, \hat{u}_{i,t}) \in \hat{\mathcal{S}}_i \times \hat{\mathcal{U}}_i$

    2  the reward function is linearly factorizable as $r(s_t, \boldsymbol{u}_t) = \sum_{i=1}^{k} r_i(\hat{s}_{i,t}, \hat{u}_{i,t})$

then we say $\mathcal{MG}$ is decomposable on $\{\hat{\mathcal{S}}_1 \times \hat{\mathcal{U}}_1, \cdots, \hat{\mathcal{S}}_k \times \hat{\mathcal{U}}_k\}$. $\{\mathcal{MG}_1, \cdots, \mathcal{MG}_k\}$ is the corresponding decomposition of $\mathcal{MG}$, where $\mathcal{MG}_i := <\hat{\mathcal{S}}_i, \hat{\mathcal{U}}_i, \mathcal{P}, r_i, n_i, \gamma> (i \in [1, k])$. $n_i$ is the number of agents involved in $\mathcal{MG}_i$. Specially, $\forall i \in [1, k]$, if $\mathcal{MG}_i$ is no longer decomposable, we say $\{\mathcal{MG}_1, \cdots, \mathcal{MG}_k\}$ is the *minimum decomposition*.

Note that the decomposition of a decomposable MMDP $\mathcal{MG}$ could be non-unique. To explore all decompositions of $\mathcal{MG}$, we propose the following property:

**Property 3.2.** *Given a decomposable MMDP* $\mathcal{MG} = <\mathcal{S}, \mathcal{U}, \mathcal{P}, r, n, \gamma>$ *and its decomposition* $\{\mathcal{MG}_1, \cdots, \mathcal{MG}_k\}$, *where* $\mathcal{MG}_i = <\hat{\mathcal{S}}_i, \hat{\mathcal{U}}_i, \mathcal{P}, r_i, n_i, \gamma> (i \in [1, k])$. *Let* $\hat{\mathcal{S}}' \times \hat{\mathcal{U}}'$ *denote a non-empty proper subset of* $\{\hat{\mathcal{S}}_1 \times \hat{\mathcal{U}}_1, \cdots, \hat{\mathcal{S}}_k \times \hat{\mathcal{U}}_k\}$. $\mathcal{MG}$ *is also decomposable on* $\{\hat{\mathcal{S}}'_1 \times \hat{\mathcal{U}}'_1 \cdots, \hat{\mathcal{S}}'_{k_s} \times \hat{\mathcal{U}}'_{k_s}\}$ *if* $\cup_{i=1}^{k_s} \hat{\mathcal{U}}'_i = \cup_{i=1}^{k} \hat{\mathcal{U}}_i$.

The proof of Property 3.2 is provided in Appendix B. Suppose $\{\mathcal{MG}_1, \cdots, \mathcal{MG}_k\}$ is the minimum decomposition of $\mathcal{MG}$, we can obtain *all decompositions* of $\mathcal{MG}$ from the all permutations of $\{\mathcal{MG}_1, \cdots, \mathcal{MG}_k\}$. For any decomposition of $\mathcal{MG}$, the following theorem holds:

**Theorem 3.3.** *Given an MMDP* $\mathcal{MG} = <\mathcal{S}, \mathcal{U}, \mathcal{P}, r, n, \gamma>$, $\forall(s, \boldsymbol{u}) \in \mathcal{S} \times \mathcal{U}$, *the action-value function is linearly factorizable as* $\mathcal{Q}(s, \boldsymbol{u}) = \sum_{i=1}^{k} \mathcal{Q}_i(\hat{s}_i, \hat{u}_i)$ ***if and only if*** $\mathcal{MG}$ *is decomposable on* $\{\hat{\mathcal{S}}_1 \times \hat{\mathcal{U}}_1, \cdots, \hat{\mathcal{S}}_k \times \hat{\mathcal{U}}_k\}$, *where* $(\hat{s}_i, \hat{u}_i) \in \hat{\mathcal{S}}_i \times \hat{\mathcal{U}}_i$ $(i \in [1, k])$.

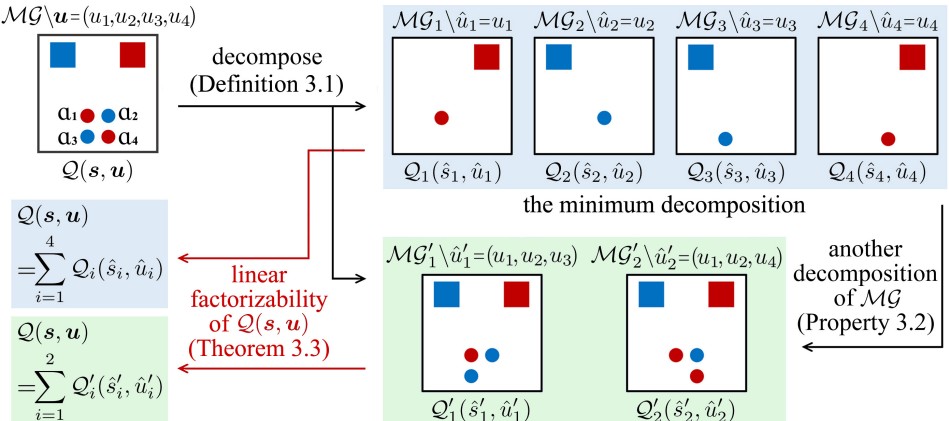

Figure 2: Examples of MMDP decomposing. 4 agents need to cover 2 landmarks, where each agent has a target landmark with the same color of it. Agents would not collide with each other. The team receives an instant reward when any agent arrives its target landmark. The task is a decomposable MMDP since: (1) the reward function is linearly factorizable to the arrival rewards of 4 agents; (2) the transition function of each agent is independent to the others'.

The proof of Theorem 3.3 is available in Appendix C. We provide intuitive differences between decomposable and indecomposable MMDPs in Appendix D. Fig.2 presents two examples of MMDP decomposing, where the minimum decomposition is marked with blue background. Theorem 3.3 indicates that if MMDP $\mathcal{MG}$ can be decomposed by a set of sub-MMDPs, the action-value functions of these sub-MMDPs are additive, which yields the action-value function of $\mathcal{MG}$.

## 3.2 DISCUSSIONS

*LMF always suffers from representational limitation in indecomposable MMDP.* There is no representational limitation for LMF if the task can be modelled by the MMDP decomposable on $\{\mathcal{S} \times \mathcal{U}_1, \cdots, \mathcal{S} \times \mathcal{U}_n\}$. In this case, the action-value function is linearly factorizable as $\mathcal{Q}(s, \boldsymbol{u}) = \sum_{a=1}^{n} \mathcal{Q}_a(s, u_a)$, which is consistent with the decomposition of the joint Q value of LMF, i.e., $Q_{tot}(s, \boldsymbol{u}) = \sum_{a=1}^{n} Q_a(s, u_a)$. Besides, LMF always suffers from representational limitation in indecomposable MMDP, where the action-value function is not linearly factorizable on any combination of the subsets of $\mathcal{S} \times \mathcal{U}$. From Definition 3.1, an MMDP involving cooperative rewards or interactive transitions of all agents is indecomposable, which is the case of most MARL tasks.

*LMF in indecomposable MMDP: unbiased estimation of Temporal Difference (TD) target under sarsa value iteration.* Consider LMF under discrete action space setting. The complete representation of action-values under state $s$ is equivalent to solving a linear equation system: $\{\mathcal{Q}(s, \boldsymbol{u}) = \sum_{a=1}^{n} Q_a(s, u_a)\}_{\forall \boldsymbol{u} \in \mathcal{U}}$. Such equation system is always *overdetermined* for indecomposable MMDP (proof is available in Appendix E), which result in $Q_{tot}(s, \boldsymbol{u}) \neq \mathcal{Q}(s, \boldsymbol{u})$, i.e., the estimation of the action-values is biased. However, the estimation of the state value and TD target are still unbiased under sarsa value iteration (proof is available in Appendix F), which suggests

the representational error would not accumulate across transitions. In this case, *an indecomposable MMDP is reducible along the trajectory into single-step matrix games*, where approaches addressing single-step matrix games are applicable to solve the optimal policy.

*Related works*. Detailed discussions of related works are available in Appendix G. The concept of multi-agent game decomposition has also been discussed by previous works (Dou et al., 2022; Castellini et al., 2021). Such works fails to figure out a reasonable relationship between game decomposition and action-value factorization. In this section we investigate LMF, the most simple case of Strictly Monotonic Mixing Function (SMMF). The SMMF proposed by QMIX (Rashid et al., 2020b) extends the expressive capacity of LMF and is wildly applied in various value decomposition methods. Such SMMF is only capable to represent the action-value functions which are also strictly monotonic. We can easily raise examples where SMMF encounters representational limitation, such as the two-agent matrix game with payoff matrix $[[1, 0], [0, 1]]$.

## 4 MIXING FUNCTIONS WITH COMPLETE REPRESENTATIONAL CAPACITY

SMMF is incapable to completely represent the correct action-values across a wild range of MARL tasks. In section 4.1, by expanding upon SMMF, we propose a two-stage mixing framework with complete representational capacity. However, the capacity could be unrealized for the representational interference on the optimal action-values, which is introduced and addressed in section 4.2.

### 4.1 MIXING FOR UNBOUNDED DIFFERENCE

Since both IGM and complete representational capacity are critical for value decomposition, we investigate the mixing function with complete representational capacity under the IGM constraint. To formalize the problem, let $R(f_\theta(x))$ denote the range of function $f$ under input $x$, i.e.,

$$R(f_\theta(x)) = \{f(x; \theta) \mid \forall \theta \in \Theta\} \tag{5}$$

where $\theta$ and $\Theta$ denote the parameter and the parameter space, respectively. We say a mixing function has complete representational capacity under IGM constraint if $Q_{tot}(s, \boldsymbol{u}_{gre}) = \mathcal{F}_\theta(\cup_{a=1}^n Q_{a,gre}, s, \boldsymbol{u}_{gre}) = \mathcal{Q}(s, \boldsymbol{u}^*)$ and

$$\forall s, \boldsymbol{u} \in \mathcal{S} \times \mathcal{U}, \ R(\mathcal{F}_\theta(\cup_{a=1}^n Q_a, s, \boldsymbol{u})) = (-\infty, \mathcal{Q}(s, \boldsymbol{u}^*)] \tag{6}$$

We only consider monotonic mixing functions since non-monotonic ones suffer from poor convergence (detailed discussion of this problem is available in Appendix H). Note that the Strictly Monotonic Mixing Function (SMMF) satisfies both IGM and monotonicity, but suffers from representational limitation. We consider completing the representational capacity of SMMF.

Suppose $f_\theta : \mathcal{R}^n \times \mathcal{S} \times \mathcal{U} \rightarrow \mathcal{R}$ is an SMMF. Let $\Delta f_\theta := f_\theta(\cup_{a=1}^n Q_{a,gre}, s, \boldsymbol{u}_{gre}) - f_\theta(\cup_{a=1}^n Q_a, s, \boldsymbol{u})$. We have $\Delta f_\theta = 0$ if $\boldsymbol{u} = \boldsymbol{u}_{gre}$, otherwise $\Delta f_\theta \geq 0$. Substituting $\Delta f_\theta$ into Eq.6, the complete representation of action-values requires

$$\forall s, \boldsymbol{u} \in \mathcal{S} \times \mathcal{U}, \ R(\Delta f_\theta) = [0, +\infty) \tag{7}$$

Given $\boldsymbol{u} \in \mathcal{U}$, there exist $\boldsymbol{u}' = \{u'_1, \cdots, u'_n\} \in \mathcal{U}$, such that $\forall a \in [1, n], Q'_a \leq Q_a \leq Q_{a,gre}$, where we abbreviate $Q_a(s, u'_a)$ as $Q'_a$. Referring to the definition of SMMF (Eq.4), we have

$$\Delta f'_\theta \geq \Delta f_\theta \geq \Delta f_{\theta,gre} \equiv 0 \tag{8}$$

where we abbreviate $\Delta f_\theta(\cup_{a=1}^n Q'_a, s, \boldsymbol{u}')$ and $\Delta f_\theta(\cup_{a=1}^n Q_{a,gre}, s, \boldsymbol{u}_{gre})$ as $\Delta f'_\theta$ and $\Delta f_{\theta,gre}$, respectively. According to Eq.8, we have $R(\Delta f_\theta(\cup_{a=1}^n Q_a, s, \boldsymbol{u})) = [0, \Delta f'] \ (0 \leq \Delta f'_\theta < +\infty)$, which violates Eq.7. Therefore, an SMMF $f$ suffers from the representational limitation due the bounded difference between the joint Q values of the greedy and other actions, i.e., $\Delta f$.

To complete the representational capacity of SMMF, we introduce a weight $w_\phi(s, \boldsymbol{u})$ and a bias $b_\psi(s, \boldsymbol{u})$ to rescale the bounded difference, where $R(w_\phi(s, \boldsymbol{u})) = [w_{min}, w_{max}]$ and $R(b_\psi(s, \boldsymbol{u})) = [b_{min}, b_{max}]$. Let

$$\Delta \mathcal{F}_{\theta,\phi,\psi}(\cup_{a=1}^n Q_a, s, \boldsymbol{u}) := w_\phi(s, \boldsymbol{u}) \cdot \Delta f_\theta + b_\psi(s, \boldsymbol{u}) \tag{9}$$

We have $R(\Delta \mathcal{F}_{\theta,\phi,\psi}(\cup_{a=1}^n Q_a, s, \boldsymbol{u})) = [b_{min}, w_{max} \cdot \Delta f'_\theta + b_{max}]$. The rescaled difference $\Delta \mathcal{F}$ is unbounded if $b_{min} = 0$ and $w_{max} + b_{max} = +\infty$. To cover a wilder range of mixing functions,

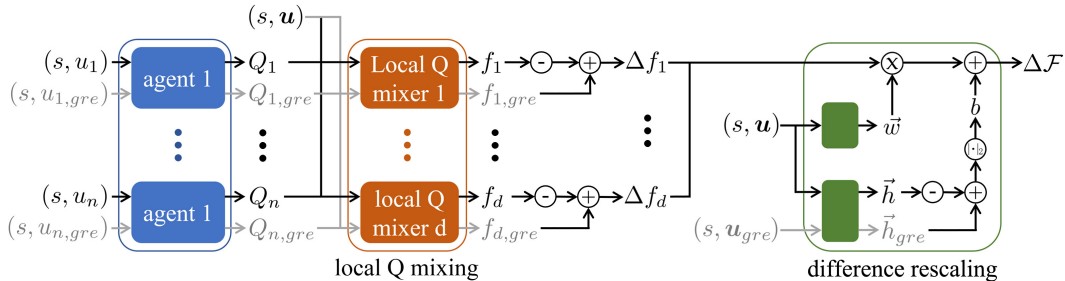

Figure 3: The pipeline of MUD. We omit the inputs and parameters. In multi-channel local Q mixing stage, each SMMF mixes the local Q values in a channel, which yields a bounded differences $\Delta f_i$ ($i \in [1, d]$). In difference rescaling stage, we mix and rescale the bounded multi-channel differences $\{\Delta f_1, \cdots, \Delta f_d\}$ into unbounded $\Delta \mathcal{F}$.

we extend $f$ to a functional vector of SMMFs, for which we have $\Delta \mathcal{F}_{\boldsymbol{\theta}, \boldsymbol{\phi}, \psi}(\cup_{a=1}^{n} Q_a, s, \boldsymbol{u}) :=$ $\vec{w}_{\boldsymbol{\phi}}(s, \boldsymbol{u}) \times \overrightarrow{\Delta f_{\boldsymbol{\theta}}} + b_\psi(s, \boldsymbol{u})$. $\vec{w}_{\boldsymbol{\phi}} = [w_{\phi,1}, \cdots, w_{\phi,d}]$ and $\overrightarrow{\Delta f_{\boldsymbol{\theta}}} = [\Delta f_{\theta,1}, \cdots, \Delta f_{\theta,d}]^\top$. $d$ is the number of SMMFs. $\times$ denotes matrix product. The IGM requires $\Delta \mathcal{F}_{\boldsymbol{\theta}, \boldsymbol{\phi}, \psi}(\cup_{a=1}^{n} Q_{a,gre}, s, \boldsymbol{u}_{gre}) \equiv$ 0. Note that $\forall i \in [1, d], \Delta f_{\theta,gre,i} \equiv 0$. We require $b_\psi(s, \boldsymbol{u}_{gre}) \equiv 0$, for which we let $b_\psi(s, \boldsymbol{u}_{gre}) = |\vec{h}(s, \boldsymbol{u}) - \vec{h}(s, \boldsymbol{u}_{gre})|_2$. Besides, since $\frac{\partial Q_{tot}(s, \boldsymbol{u})}{\partial Q_a} = \sum_{i=1}^{d} w_{\phi,i}(s, \boldsymbol{u}) \cdot \frac{\partial f}{\partial Q_a}$ and $\frac{\partial f}{\partial Q_a} \geq 0$, $\mathcal{F}$ is monotonic if we let $w_{\phi,i}(s, \boldsymbol{u}) \geq 0, \forall i \in [1, d]$.

We name this framework Mixing for Unbounded Difference (MUD). The pipeline of MUD is shown in Fig.3. The joint Q value function of MUD equals

$$Q_{tot}(s, \boldsymbol{u}) = \mathcal{F}_{\boldsymbol{\theta}, \boldsymbol{\phi}, \psi}(\cup_{a=1}^{n} Q_a, s, \boldsymbol{u}) = V(s) - \vec{w}_{\boldsymbol{\phi}}(s, \boldsymbol{u}) \times \overrightarrow{\Delta f_{\boldsymbol{\theta}}} - b_\psi(s, \boldsymbol{u}) \tag{10}$$

Eq.10 degenerates to QPLEX (Wang et al., 2020b) when (1) $w_{min} = 0$ and $w_{max} = +\infty$; (2) $b_{max} = 0$ (i.e., $b_\psi(s, \boldsymbol{u}) \equiv 0$); (3) $\overrightarrow{\Delta f_{\boldsymbol{\theta}}} = [\Delta f_{\theta,1}, \cdots, \Delta f_{\theta,n}]^\top$, where $f_{\theta,a}(\cup_{a=1}^{n} Q_a, s, \boldsymbol{u}) = Q_a$ ($a \in [1, n]$); (4) $V(s) = \sum_{a=1}^{n} Q_{a,gre}$.

## 4.2 ADDRESSING OPTIMAL REPRESENTATIONAL INTERFERENCE IN MUD

While MUD achieve complete representational capacity formally, it does not imply the correct action-values can be represented completely. It also depends on the training. Such capacity could remain unrealized due to optimal representational interference.

As shown in Fig.4, the training of $Q_{tot}(s, 1, 1)$ shapes $Q_1(s, 1)$ by grad 1 $= \mathcal{F}'_1(s, 1, 1)$, where $Q_{tot}(s, u_1, u_2) = \mathcal{F}(Q_1(s, u_1), Q_2(s, u_2), s, \boldsymbol{u})$. Since $Q_1(s, 1)$ is involved in the mixing of $Q_{tot}(s, 1, 2)$, the training of $Q_{tot}(s, 1, 1)$ affects $Q_{tot}(s, 1, 2)$. Similarly, the training of $Q_{tot}(s, 1, 2)$ also shapes $Q_1(s, 1)$, which affects the training of $Q_{tot}(s, 1, 1)$. Therefore, The training of $Q_{tot}(s, 1, 1)$ and $Q_{tot}(s, 1, 2)$ affects each other by their shared local Q value $Q_1(s, 1)$. We name this problem as *representational cross interference*. The gradient $\mathcal{F}'_1(s, 1, 1)$ determines how much the representation of $\mathcal{Q}(s, 1, 1)$ shapes $Q_1(s, 1)$, which can be viewed as its representational weight. For example, if $\mathcal{F}'_1(s, 1, 1) = 1$ while $\mathcal{F}'_1(s, 1, 2) = 10^{-6}$, the gradient for the representation of $\mathcal{Q}(s, 1, 2)$ would be overwhelmed by the interference from the representation of $\mathcal{Q}(s, 1, 1)$.

Formally, $\forall \boldsymbol{u}, \boldsymbol{u}' \in \mathcal{U}$, if $\boldsymbol{u} \cap \boldsymbol{u}' \neq \emptyset$, we say the representations of $\mathcal{Q}(s, \boldsymbol{u})$ and $\mathcal{Q}(s, \boldsymbol{u}')$ suffer from cross interference. The interference to the representation of the optimal action-value (*Optimal Representational Interference (ORI)* for brevity) is a noteworthy issue. Due to the IGM constraint, the joint Q value function of MUD is only capable to represent the action-values lower than the greedy. In this case, the complete

Figure 4: Example of representational cross interference. $u_a \in \{1, 2\}$ ($a \in \{1, 2\}$). The trainings of $Q_{tot}(s, 1, 1)$ and $Q_{tot}(s, 1, 2)$ affect each other by shaping the shared local Q value $Q_1(s, 1)$.

representation of action-values requires: the greedy action is also the optimal one. Therefore, a well representation of the optimal action-value is critical, which would fail for the ORI. To evaluate the ORI in MUD, we define the relative representational weight of the optimal action-value as *optimal representation ratio*, which is denoted by

$$\boldsymbol{w}^* = \frac{1}{n}\sum_{a=1}^n w_a^*, \quad where \quad w_a^* = \frac{\pi(\boldsymbol{u}|s)\cdot\mathcal{F}_a'(s,\boldsymbol{u}^*)}{\sum_{u_{\backslash a}}^{\mathcal{U}_{\backslash a}}\pi_a(u_a^*,u_{\backslash a}|s)\cdot\mathcal{F}_a'(s,u_a^*,u_{\backslash a})} \tag{11}$$

where $\mathcal{F}_a'(s,\boldsymbol{u}) = \frac{\partial Q_{tot}(s,\boldsymbol{u})}{\partial Q_a(s,u_a)}$. An example is provided in Appendix I, where the optimal representation ratio declines to almost 0 during training. As a result, the joint Q value function suffers from poor convergence due to severe ORI.

To address the ORI, we shape the gradient to improve the representation weight of the optimal action-value. Note that the mixed difference $\Delta\mathcal{F}$ can be viewed as the evaluated performance of current action. The action is better if $\Delta\mathcal{F}$ is smaller. Therefore, we consider reducing the gradient according to the value of $\Delta\mathcal{F}$ to improve the optimal representational weight. Consider a basic form of difference mixing in Eq.9. We replace the weight $w_\phi(s,\boldsymbol{u})$ with an exponentially decaying multiplier $e^{-b_\psi(s,\boldsymbol{u})}$. Let $f_\theta(\cup_{a=1}^n Q_a, s, \boldsymbol{u}) = \frac{1}{n}\sum_{a=1}^n\left(1 - \frac{Q_a}{Q_{a,gre}}\right)$. We have

$$\Delta\mathcal{F}_\psi(\cup_{a=1}^n Q_a, s, \boldsymbol{u}) = e^{-b_\psi(s,\boldsymbol{u})}\cdot\frac{1}{n}\sum_{a=1}^n\left(1 - \frac{Q_a}{Q_{a,gre}}\right) + b_\psi(s,\boldsymbol{u}) \tag{12}$$

Eq.12 is named MUD with Smooth Gradients (MUD-SmG). The representation weight equals $\frac{\partial\mathcal{F}}{\partial Q_a} = \frac{1}{n}e^{-b_\psi(s,\boldsymbol{u})}$. Note that $\frac{\partial\Delta\mathcal{F}}{\partial b_\psi(s,\boldsymbol{u})} = 1 - \frac{1}{n}\sum_{a=1}^n\left(1 - \frac{Q_a}{Q_{a,gre}}\right)e^{-b_\psi(s,\boldsymbol{u})} > 0$. As a result, the representation weight decreases monotonically with $\Delta\mathcal{F}$.

Eq.12 can be further simplified by replacing $e^{-b_\psi(s,\boldsymbol{u})}$ with a 1-0 step function $w(b_\psi)$, where $w(b_\psi) = 1$ for $b_\psi(s,\boldsymbol{u}) < \alpha$, otherwise $w(b_\psi) = 0$. $\alpha$ is a hyper-parameter. We have

$$\Delta\mathcal{F}_{\boldsymbol{\theta},\boldsymbol{\phi},\psi}(\cup_{a=1}^n Q_a, s, \boldsymbol{u}) = \begin{cases} \sum_{a=1}^n\left(Q_{a,gre} - SG(Q_a)\right) + b_\psi(s,\boldsymbol{u}) & b_\psi(s,\boldsymbol{u}) > \alpha \\ \sum_{a=1}^n\left(Q_{a,gre} - Q_a\right) + b_\psi(s,\boldsymbol{u}) & otherwise \end{cases} \tag{13}$$

where $SG$ denotes stopping gradient. Eq.13 is named MUD with Stepped Gradients (MUD-StG). Both MUD-SmG and MUD-StG alleviate the ORI by reducing the representation weight of action-values with high mixed difference $\Delta\mathcal{F}$. We will discuss the effects of both mixing functions in the experimental parts. Related works can be found in Appendix G.

## 5 EXPERIMENTS

Our experiments consist of 3 parts: (1) Verification of the expressiveness of Linear Mixing Function (LMF); (2) Evaluation of our proposed mixing functions; (3) Ablation studies.

### 5.1 VERIFICATION OF THE EXPRESSIVENESS OF LMF

Referring to Fig.2, we design toy games to verify the expressiveness of LMF. As shown in Fig.5. The map is gridded by a $4\times 4$ checkerboard. All agents are initialized with the position $(3,0)$ and required to select actions from $\{up, right\}$ at each time step. The team receives an instant reward of 1 when any agent arrives and gets accepted by a landmark. Each landmark only accepts the agent with the same color as it and is limited to accept *no more than* 2 agents. Invalid action, e.g, $up$ at position $(0,0)$ are masked. Detailed tasks and training setups are available in Appendix J.1.

For the decomposable case, consider two decompositions of $\mathcal{MG}$ shown in Fig.2. According to Theorem 3.3, the action-value function is linearly factorizable as $\mathcal{Q}(s,\boldsymbol{u}) = \sum_{a=1}^n\mathcal{Q}_a(s,u_a)$ and $\mathcal{Q}(s,\boldsymbol{u}) =$

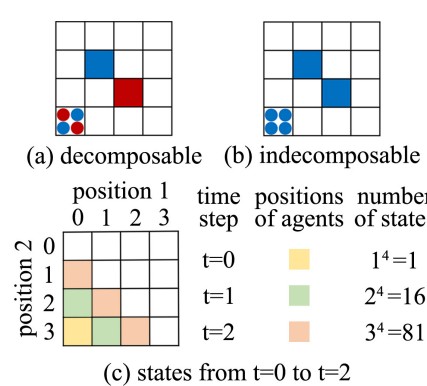

(a) decomposable  (b) indecomposable

(c) states from t=0 to t=2

Figure 5: Setup of the toy game.

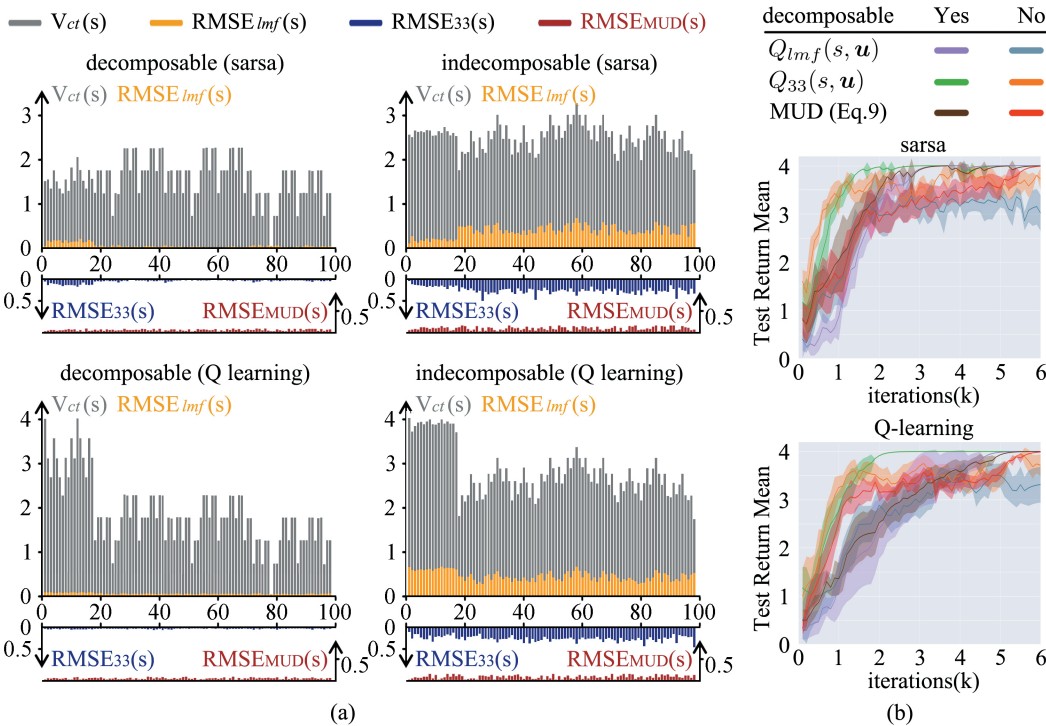

Figure 6: Verification of the expressiveness of LMF. (a) Test RMSE; (b) Test mean return.

$\mathcal{Q}'_1(s, u_1, u_2, u_3) + \mathcal{Q}'_2(s, u_1, u_2, u_4)$ for these two decompositions, respectively. We apply neural networks to model two joint Q value functions: (1) $Q_{lmf}(s, \boldsymbol{u}) = \sum_{a=1}^n Q_a(s, u_a)$; (2) $Q_{33}(s, \boldsymbol{u}) = \mathcal{Q}'_1(s, u_1, u_2, u_3) + \mathcal{Q}'_2(s, u_1, u_2, u_4)$. We test the Root Mean Square representational Errors (RMSE) of both joint Q value functions. The actual action-values are approximated by a non-factorized Q value function $Q_{ct}(s, \boldsymbol{u})$

After 6k iterations of training, $RMSE_{33}(s)$ and $RMSE_{lmf}(s)$ are tested for the 98 states of the first 3 time steps. The results are shown in Fig.6(a), where each bar denotes the a single state. Note that both $RMSE_{33}(s)$ and $RMSE_{lmf}(s)$ are negligible in decomposable case. In words, both $Q_{lmf}(s, \boldsymbol{u})$ and $Q_{33}(s, \boldsymbol{u})$ are able to represent $\mathcal{Q}(s, \boldsymbol{u})$ (with negligible errors), which suggests $\mathcal{Q}(s, \boldsymbol{u})$ is linearly factorizable in decomposable case. By contrast, we can infer that $\mathcal{Q}(s, \boldsymbol{u})$ is not linearly factorizable in indecomposable case since none of $Q_{lmf}(s, \boldsymbol{u})$ and $Q_{33}(s, \boldsymbol{u})$ is able to represent $\mathcal{Q}(s, \boldsymbol{u})$ concisely. Fig.6(b) presents the test return during training, where both $Q_{lmf}(s, \boldsymbol{u})$ and $Q_{33}(s, \boldsymbol{u})$ achieve the largest return of 4 in the decomposable case but receive returns less than 4 in indecomposable case. In words, the linearly factorized joint Q value function is capable to solve the decomposable MMDP but fails in the indecomposable MMDP.

We also evaluate the expressiveness of MUD (Eq.9). The experimental results indicate that MUD is capable to completely represent the action-values in both decomposable and indecomposable MMDPs. As shown in Fig.6(b), MUD achieves the highest return in both cases.

## 5.2 EVALUATION OF MIXING FUNCTIONS

Firstly, we evaluate the performance of different mixing functions in a 2-agent single-step matrix game, where each agent chooses an action from $\{1, 2, 3\}$ and receives a team reward according to the payoff matrix in Fig.7. Detailed training setups are available in Appendix J.2. The mean representational errors do not decrease for LMF and SMMF for the representational limitation. Both LMF and SMMF are easily stuck in the sub-optimal point with the return of 6. By contrast, the mean representational errors decline quickly for the mixing functions with complete representational capability. However, since the optimal representational error remains high for declining optimal representational ratio $\boldsymbol{w}^*$, such capacity is unrealized for MUD (QPLEX), MUD(QPLEX-SG), and

MUD (Eq.9). As a result, these methods are still easily stuck in the sub-optimal point. Both of our proposed mixing functions (i.e., MUD-SmG and MUD-StG) overcome the ORI and jump out of the sub-optimum. We can see $w^*$ of MUD-SmG and MUD-StG grows and approximates 1.

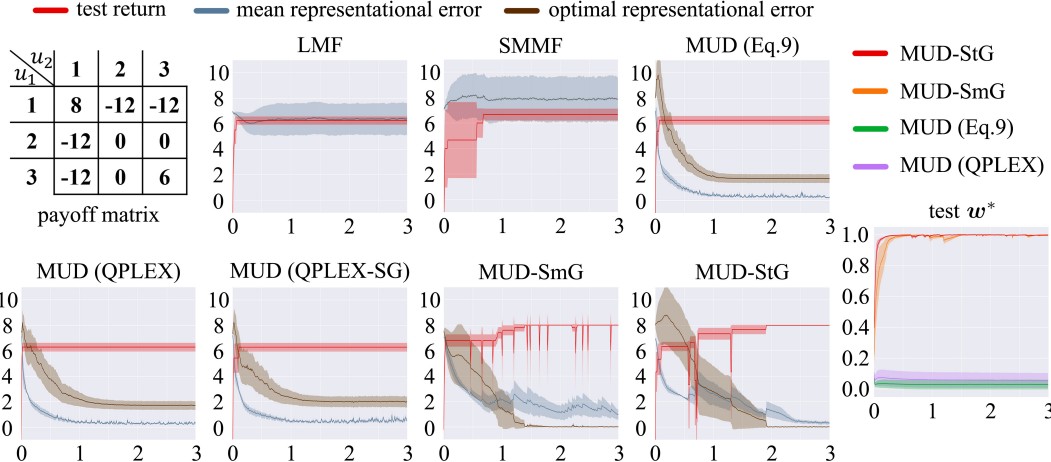

Figure 7: Evaluation of mixing functions on matrix game. The expressions of the involved mixing functions are collected and presented in Appendix J.2. The optimal representational error refers to the representational error of the optimal action-value. $w^* \in [0, 1]$ (Eq.11) is a qualification index of ORI. The $w^*$ of LMF, SMMF, and MUD (QPLEX-SG) are constants, which are not presented here.

To evaluate the scalability of propose mixing functions, we exam MUD-SmG and MUD-StG in predator prey environments under partial observation. As shown in Fig.8, the direct implementations of LMF (i.e., VDN) and SMMF (i.e., QMIX) fail in both tasks. By contrast, QTRAN (implementation of LMF) and WQMIX (implementation of SMMF) focus on the representation of underestimated action-values, which alleviates the ORI. Therefore, both QTRAN and WQMIX are able to handle the task with mild punishments. QPLEX has complete representational capability but still fails in all tasks due to the ORI. Only MUD-StG and MUD-SmG solve the task under the punishment of -5.

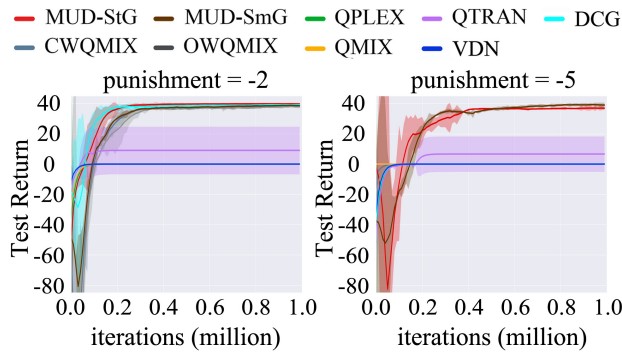

Figure 8: Evaluation of value decomposition methods in predator prey with the punishments of -2 and -5.

For additional experimental results, please consult Appendix K.

## 6 CONCLUSIONS

In this paper, we investigate the problem of representational limitation from the perspectives of task property and mixing function, which yields the following findings: (1) LMF is free from representational limitation only in a rare case of MARL problems, namely decomposable MMDP. The action-value function of decomposable MMDP is linearly factorizable into the action-value function of sub-MMDPs. (2) SMMF suffers from representational limitation for the bounded differences between the joint Q values of the greedy and other actions. We propose the framework of MUD, which overcomes the limitation by mapping the differences into unbound variables. (3) The representation of action-values encounters cross interferences for value decomposition. The cross interference to the representation of the optimal action-values (i.e., ORI) would bring convergence difficulties. The experimental results validate the expressiveness of LMF, reveals the importance of ORI, and demonstrate the effectiveness of our methods to address the problem.

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

## A  STRICTLY MONOTONIC MIXING FUNCTIONS SATISFY THE IGM

Suppose $\mathcal{F}$ is strictly monotonic and $Q_{tot}(s, \boldsymbol{\tau}, \boldsymbol{u}) = \mathcal{F}\left(\cup_{a=1}^{n} Q_a(\tau_a, u_a), s, \boldsymbol{\tau}, \boldsymbol{u}\right)$. Here we prove that $\mathcal{F}$ satisfies the IGM.

*Proof.* Note that the local greedy action is defined as $u_{a,gre} := \arg\max_{u_a} Q_a(\tau_a, u_a)$. We have $\forall u_a \in \mathcal{U}_a$, $Q_a(\tau_a, u_{a,gre}) > Q_a(\tau_a, u_a)$. Referring to the definition of the strictly monotonic mixing function (Eq.4), since $\mathcal{F}$ is strictly monotonic, we have $\forall u_a \in \mathcal{U}_a$,

$$Q_{tot}(s, \boldsymbol{\tau}, u_{a,gre}, u_{\backslash a}) > Q_{tot}(s, \boldsymbol{\tau}, u_a, u_{\backslash a}) \tag{14}$$

where $u_{\backslash a}$ denotes the set of actions except $u_a$, i.e., $u_{\backslash a} \cup u_a = \boldsymbol{u}$. It can be inferred from Eq.14 that $\forall \boldsymbol{u} = \{u_1, \cdots, u_n\} \in \boldsymbol{\mathcal{U}}$,

$$Q_{tot}(s, \boldsymbol{\tau}, u_{1,gre}, \cdots, u_{n,gre}) > Q_{tot}(s, \boldsymbol{\tau}, u_1, \cdots, u_n) \tag{15}$$

Therefore, we have $\arg\max_{\boldsymbol{u}} Q_{tot}(s, \boldsymbol{\tau}, \boldsymbol{u}) = \boldsymbol{u}_{gre} = \{\arg\max_{u_1} Q_1(\tau_1, u_1), \cdots, \arg\max_{u_n} Q_n(\tau_n, u_n)\}$. According to the definition of IGM (Eq.2), $\mathcal{F}$ satisfies the IGM. $\square$

## B  PROOF OF PROPERTY 3.2

Given a decomposable MMDP $\mathcal{MG} = <\mathcal{S}, \boldsymbol{\mathcal{U}}, \mathcal{P}, r, n, \gamma>$ and its decomposition $\{\mathcal{MG}_1, \cdots, \mathcal{MG}_k\}$, where $\mathcal{MG}_i = <\hat{\mathcal{S}}_i, \hat{\mathcal{U}}_i, \mathcal{P}, r_i, n_i, \gamma> (i \in [1, k])$. Let $\hat{\mathcal{S}}' \times \hat{\mathcal{U}}'$ denote a non-empty proper subset of $\{\hat{\mathcal{S}}_1 \times \hat{\mathcal{U}}_1, \cdots, \hat{\mathcal{S}}_k \times \hat{\mathcal{U}}_k\}$. Assume $\cup_{i=1}^{k_s} \hat{\mathcal{U}}_i' = \cup_{i=1}^{k} \hat{\mathcal{U}}_i$. Here we prove that $\mathcal{MG}$ is decomposable on $\{\hat{\mathcal{S}}_1' \times \hat{\mathcal{U}}_1', \cdots, \hat{\mathcal{S}}_{k_s}' \times \hat{\mathcal{U}}_{k_s}'\}$.

*Proof.* Let $I_{i,j} = 1$ if $\hat{\mathcal{S}}_i \times \hat{\mathcal{U}}_i \subset \hat{\mathcal{S}}'_j \times \hat{\mathcal{U}}'_j$ ($i \in [1,k]$, $j \in [1,k_s]$), otherwise $I_{i,j} = 0$. We have $\hat{\mathcal{U}}'_j = \cup_{i=1}^k I_{i,j} \cdot \hat{\mathcal{U}}_i$ and $\hat{\mathcal{S}}'_j = \cup_{i=1}^k I_{i,j} \cdot \hat{\mathcal{S}}_i$. Note that $\cup_{i=1}^{k_s} \hat{\mathcal{U}}'_i = \cup_{i=1}^k \hat{\mathcal{U}}_i$. We have

$$
\begin{aligned}
\cup_{j=1}^{k_s} \hat{\mathcal{U}}'_j &= \cup_{j=1}^{k_s} \left( \cup_{i=1}^k I_{i,j} \cdot \hat{\mathcal{U}}_i \right) \\
&= \cup_{j=1}^{k_s} I_{i,j} \cdot \cup_{i=1}^k \hat{\mathcal{U}}_i = \cup_{i=1}^k \hat{\mathcal{U}}_i
\end{aligned}
\tag{16}
$$

which indicates $\forall j \in [1,k]$, $\sum_{j=1}^{k_s} I_{i,j} \geq 1$.

Let $r'_j(\hat{s}'_j, \hat{u}'_j)$ denote a reward function defined on $\hat{\mathcal{S}}'_j \times \hat{\mathcal{U}}'_j$, which equals

$$
r'_j(\hat{s}'_j, \hat{u}'_j) := \sum_{i=1}^k \frac{I_{i,j} \cdot r_i(\hat{s}_i, \hat{u}_i)}{\sum_{j=1}^{k_s} I_{i,j}}
\tag{17}
$$

The sum of all reward functions defined on $\{\hat{\mathcal{S}}'_1 \times \hat{\mathcal{U}}'_1, \cdots, \hat{\mathcal{S}}'_{k_s} \times \hat{\mathcal{U}}'_{k_s}\}$ equals

$$
\begin{aligned}
\sum_{j=1}^{k_s} r'_j(\hat{s}'_j, \hat{u}'_j) = \sum_{j=1}^{k_s} \sum_{i=1}^k \frac{I_{i,j} \cdot r_i(\hat{s}_i, \hat{u}_i)}{\sum_{j=1}^{k_s} I_{i,j}} &= \sum_{i=1}^k \left( \sum_{j=1}^{k_s} \frac{I_{i,j} \cdot r_i(\hat{s}_i, \hat{u}_i)}{\sum_{j=1}^{k_s} I_{i,j}} \right) \\
&= \sum_{i=1}^k \frac{\left( \sum_{j=1}^{k_s} I_{i,j} \right) \cdot r_i(\hat{s}_i, \hat{u}_i)}{\sum_{j=1}^{k_s} I_{i,j}} = \sum_{i=1}^k r(\hat{s}_i, \hat{u}_i) = r(s, \boldsymbol{u})
\end{aligned}
\tag{18}
$$

We have proved that the reward function $r(s, \boldsymbol{u})$ is linearly factorizable on $\{\hat{\mathcal{S}}'_1 \times \hat{\mathcal{U}}'_1, \cdots, \hat{\mathcal{S}}'_{k_s} \times \hat{\mathcal{U}}'_{k_s}\}$.

Since $\{\mathcal{MG}_1, \mathcal{MG}_2, \cdots, \mathcal{MG}_k\}$ is a decomposition of $\mathcal{MG}$, we have $\forall i \in [1,k]$, $\mathcal{P}(\hat{s}_{i,t+1}|\hat{s}_{i,t}, \hat{u}_{i,t}) = \mathcal{P}(\hat{s}_{i,t+1}|s_t, \boldsymbol{u}_t)$. Note that $\hat{\mathcal{U}}'_j = \cup_{i=1}^k I_{i,j} \cdot \hat{\mathcal{U}}_i$ and $\hat{\mathcal{S}}'_j = \cup_{i=1}^k I_{i,j} \cdot \hat{\mathcal{S}}_i$. We have $\hat{s}'_j = \cup_{i=1}^k I_{i,j} \cdot \hat{s}_i$ and $\hat{u}'_j = \cup_{i=1}^k I_{i,j} \cdot \hat{u}_i$. Therefore,

$$
\begin{aligned}
\mathcal{P}(\hat{s}'_{j,t+1}|\hat{s}'_{j,t}, \hat{u}'_{j,t}) &= \mathcal{P}(\cup_{i=1}^k I_{i,j} \cdot \hat{s}_{i,t+1}| \cup_{i=1}^k I_{i,j} \cdot \hat{s}_{i,t}, \cup_{i=1}^k I_{i,j} \cdot \hat{u}_{i,t}) \\
&= \mathcal{P}(\cup_{i=1}^k I_{i,j} \cdot \hat{s}_{i,t+1}|s_t, \boldsymbol{u}_t) = \mathcal{P}(\hat{s}'_{j,t+1}|s_t, \boldsymbol{u}_t)
\end{aligned}
\tag{19}
$$

According to Definition 3.1, $\mathcal{MG}$ is decomposable on $\{\hat{\mathcal{S}}'_1 \times \hat{\mathcal{U}}'_1, \cdots, \hat{\mathcal{S}}'_{k_s} \times \hat{\mathcal{U}}'_{k_s}\}$. $\square$

## C    PROOF OF THEOREM 3.3

### C.1    PROOF OF SUFFICIENCY

Given a MMDP $\mathcal{MG} = < \mathcal{S}, \boldsymbol{\mathcal{U}}, \mathcal{P}, r, n, \gamma >$. $\mathcal{MG}$ is decomposable on $\{\hat{\mathcal{S}}_1 \times \hat{\mathcal{U}}_1, \cdots, \hat{\mathcal{S}}_k \times \hat{\mathcal{U}}_k\}$, where $\forall i, j \in [1,k], \hat{\mathcal{U}}_i \neq \hat{\mathcal{U}}_j$ if $i \neq j$. $\forall i \in [1,k]$, $\hat{\mathcal{S}}_i$ and $\hat{\mathcal{U}}_i$ denote a subset of the state space and a non-empty proper subset of the joint action space, respectively, i.e., $\hat{\mathcal{S}}_i \subset \mathcal{S}, \hat{\mathcal{U}}_i \subsetneq \boldsymbol{\mathcal{U}}$ and $\hat{\mathcal{U}}_i \neq \emptyset$. Here we prove that the action-value function is linearly factorizable as $\mathcal{Q}(s, \boldsymbol{u}) = \sum_{i=1}^k \mathcal{Q}_i(\hat{s}_i, \hat{u}_i)$ under both sarsa and Q-learning value iteration, where $(s, \boldsymbol{u}) \in \mathcal{S} \times \boldsymbol{\mathcal{U}}$ and $(\hat{s}_i, \hat{u}_i) \in \hat{\mathcal{S}}_i \times \hat{\mathcal{U}}_i$ ($i \in [1,k]$).

*Proof.* Firstly, consider the action-value function under the sarsa value iteration. Assume $\mathcal{Q}(s_{t+1}, \boldsymbol{u}_{t+1})$ is linearly factorizable as

$$
\mathcal{Q}(s_{t+1}, \boldsymbol{u}_{t+1}) = \sum_{i=1}^k \mathcal{Q}_i(\hat{s}_{i,t+1}, \hat{u}_{i,t+1})
\tag{20}
$$

Note the environment is fully observable. We have $\mathcal{Q}_i(\hat{s}_{i,t+1}, \hat{u}_{i,t+1}) = \mathcal{Q}_i(s_{t+1}, \hat{u}_{i,t+1})$. The state value function of $s_{t+1}$ equals

$$
V(s_{t+1}) = \int_{\boldsymbol{u}_{t+1}} \pi(\boldsymbol{u}_{t+1}|s_{t+1}) \mathcal{Q}(s_{t+1}, \boldsymbol{u}_{t+1}) d\boldsymbol{u}_{t+1} = \sum_{i=1}^k \int_{\boldsymbol{u}_{t+1}} \pi(\boldsymbol{u}_{t+1}|s_{t+1}) \mathcal{Q}_i(\hat{s}_{i,t+1}, \hat{u}_{i,t+1}) d\boldsymbol{u}_{t+1}
\tag{21}
$$

Since the local policies are decentralized, they are independent of each other. Let $\hat{\pi}_i(\hat{u}_{i,t+1}|s_{t+1})$ denote the joint policy of the agents involved in $\mathcal{MG}_i$. We have

$$\int_{\boldsymbol{u}_{t+1}} \pi(\boldsymbol{u}_{t+1}|s_{t+1})\mathcal{Q}_i(\hat{s}_{i,t+1}, \hat{u}_{i,t+1})d\boldsymbol{u}_{t+1} = \int_{\hat{u}_{i,t+1}} \hat{\pi}_i(\hat{u}_{i,t+1}|s_{t+1})\mathcal{Q}_i(\hat{s}_{i,t+1}, \hat{u}_{i,t+1})d\hat{u}_{i,t+1}$$
(22)

The policy $\hat{\pi}_i(\hat{u}_{i,t+1}|s_{t+1})$ is generally defined and determined by $\mathcal{Q}_i(s_{t+1}, \hat{u}_{i,t+1})$ for on-policy data distribution, e.g., $\epsilon$-greedy policy or multinomial distribution policy. Note that $\mathcal{Q}_i(s_{t+1}, \hat{u}_{i,t+1}) = \mathcal{Q}_i(\hat{s}_{i,t+1}, \hat{u}_{i,t+1})$, which indicates $\hat{\pi}_i(\hat{u}_{i,t+1}|s_{t+1}) = \hat{\pi}_i(\hat{u}_{i,t+1}|\hat{s}_{i,t+1})$. Substituting Eq.22 into Eq.21, we have

$$V(s_{t+1}) = \sum_{i=1}^{k} \int_{\hat{u}_{i,t+1}} \hat{\pi}_i(\hat{u}_{i,t+1}|\hat{s}_{i,t+1})\mathcal{Q}_i(\hat{s}_{i,t+1}, \hat{u}_{i,t+1})d\hat{u}_{i,t+1}$$
(23)

Let $V_i(\hat{s}_{i,t+1}) := \int_{\hat{u}_{i,t+1}} \hat{\pi}_i(\hat{u}_{i,t+1}|\hat{s}_{i,t+1})\mathcal{Q}_i(\hat{s}_{i,t+1}, \hat{u}_{i,t+1})d\hat{u}_{i,t+1}$ denote the state value function of $\mathcal{MG}_i$. We have $V(s_{t+1}) = \sum_{i=1}^{k} V_i(\hat{s}_{i,t+1})$.

Since $\mathcal{MG}$ is decomposable, the reward function is linearly factorizable as $r(s_t, \boldsymbol{u}_t) = \sum_{i=1}^{k} r_i(\hat{s}_{i,t}, \hat{u}_{i,t})$. The action-value function at time step $t$ equals

$$\mathcal{Q}(s_t, \boldsymbol{u}_t) = r(s_t, \boldsymbol{u}_t) + \gamma \int_{s_{t+1}} \mathcal{P}(s_{t+1}|s_t, \boldsymbol{u}_t)V(s_{t+1})ds_{t+1}$$
$$= \sum_{i=1}^{k} r_i(\hat{s}_{i,t}, \hat{u}_{i,t}) + \gamma \sum_{i=1}^{k} \int_{s_{t+1}} \mathcal{P}(s_{t+1}|s_t, \boldsymbol{u}_t)V_i(\hat{s}_{i,t+1})ds_{t+1}$$
(24)

Note that $\mathcal{S}_i$ is a subset of $\mathcal{S}$. We have

$$\int_{s_{t+1}} \mathcal{P}(s_{t+1}|s_t, \boldsymbol{u}_t)V_i(\hat{s}_{i,t+1})ds_{t+1} = \int_{\hat{s}_{i,t+1}} \mathcal{P}(\hat{s}_{i,t+1}|s_t, \boldsymbol{u}_t)V_i(\hat{s}_{i,t+1})d\hat{s}_{i,t+1}$$
(25)

Since $\mathcal{MG}$ is decomposable, we have $P(\hat{s}_{i,t+1}|\hat{s}_{i,t}, \hat{u}_{i,t}) = P(\hat{s}_{i,t+1}|s_t, \boldsymbol{u}_t)$. Substituting this equality and Eq.25 into Eq.24, we have

$$\mathcal{Q}(s_t, \boldsymbol{u}_t) = \sum_{i=1}^{k} \left( r_i(\hat{s}_{i,t}, \hat{u}_{i,t}) + \gamma \int_{\hat{s}_{i,t+1}} \mathcal{P}(\hat{s}_{i,t+1}|\hat{s}_{i,t}, \hat{u}_{i,t})V_i(\hat{s}_{i,t})d\hat{s}_{i,t+1} \right) = \sum_{i=1}^{k} \mathcal{Q}_i(\hat{s}_{i,t}, \hat{u}_{i,t})$$
(26)

where $\mathcal{Q}_i(\hat{s}_{i,t}, \hat{u}_{i,t}) := r_i(\hat{s}_{i,t}, \hat{u}_{i,t}) + \gamma \int_{\hat{s}_{i,t+1}} \mathcal{P}(\hat{s}_{i,t+1}|\hat{s}_{i,t+1}, \hat{u}_{i,t+1})V_i(\hat{s}_{i,t+1})d\hat{s}_{i,t+1}$ is defined as the action-state value function of $\mathcal{MG}_i$.

For the joint Q value function under Q-learning value iteration, assume $\mathcal{Q}(s_{t+1}, \boldsymbol{u}_{t+1})$ is linearly factorizable as Eq.20. We have

$$\max_{\boldsymbol{u}_{t+1}\in\mathcal{U}} \mathcal{Q}(s_{t+1}, \boldsymbol{u}_{t+1}) = \sum_{i=1}^{k} \max_{\hat{u}_{i,t+1}\in\hat{\mathcal{U}}_i} \mathcal{Q}_i(\hat{s}_{i,t+1}, \hat{u}_{i,t+1})$$
(27)

Referring to the deduction of the sarsa case, we have

$$\mathcal{Q}(s_t, \boldsymbol{u}_t) = r(s_t, \boldsymbol{u}_t) + \gamma \int_{s_{t+1}} \mathcal{P}(s_{t+1}|s_t, \boldsymbol{u}_t) \cdot \max_{\boldsymbol{u}_{t+1}\in\mathcal{U}} \mathcal{Q}(s_{t+1}, \boldsymbol{u}_{t+1})ds_{t+1}$$
$$= \sum_{i=1}^{k} \left( r_i(\hat{s}_{i,t}, \hat{u}_{i,t}) + \gamma \int_{s_{t+1}} \mathcal{P}(s_{t+1}|s_t, \boldsymbol{u}_t) \cdot \max_{\hat{u}_{i,t+1}\in\hat{\mathcal{U}}_i} \mathcal{Q}_i(\hat{s}_{i,t+1}, \hat{u}_{i,t+1})ds_{t+1} \right)$$
$$= \sum_{i=1}^{k} \left( r_i(\hat{s}_{i,t}, \hat{u}_{i,t}) + \gamma \int_{\hat{s}_{i,t+1}} \mathcal{P}(\hat{s}_{i,t+1}|\hat{s}_{i,t}, \hat{u}_{i,t}) \max_{\hat{u}_{i,t+1}\in\hat{\mathcal{U}}_i} \mathcal{Q}_i(\hat{s}_{i,t+1}, \hat{u}_{i,t+1})d\hat{s}_{i,t+1} \right)$$
$$= \sum_{i=1}^{k} \mathcal{Q}_i(\hat{s}_{i,t}, \hat{u}_{i,t})$$
(28)

We have proved that if $\mathcal{Q}(s_{t+1}, \boldsymbol{u}_{t+1})$ is linearly factorizable, $\mathcal{Q}(s_t, \boldsymbol{u}_t)$ is also linearly factorizable. For a finite MMDP, let $\mathcal{Q}(s_T, \boldsymbol{u}_T) = \mathcal{Q}_i(\hat{s}_{i,T}, \hat{u}_{i,T}) = 0 \ \forall i \in [1, k]$, where $T$ is the terminal time step. Note that $\mathcal{Q}(s_T, \boldsymbol{u}_T)$ is linearly factorizable as $\mathcal{Q}(s_T, \boldsymbol{u}_T) = \sum_{i=1}^{k} \mathcal{Q}_i(\hat{s}_{i,T}, \hat{u}_{i,T}) = 0$. Therefore, $\mathcal{Q}(s_t, \boldsymbol{u}_t)$ is linearly factorizable $\forall t \in [0, T]$. $\qquad\square$

## C.2 PROOF OF NECESSITY

Given an MMDP $\mathcal{MG} = \ <\mathcal{S}, \boldsymbol{\mathcal{U}}, \mathcal{P}, r, n, \gamma>$, assume the action-value function $\mathcal{Q}(s_t, \boldsymbol{u}_t)$ is linearly factorizable as $\mathcal{Q}(s_t, \boldsymbol{u}_t) = \sum_{i=1}^{k} \mathcal{Q}_i(\hat{s}_{i,t}, \hat{u}_{i,t})$, where $(s, \boldsymbol{u}) \in \mathcal{S} \times \boldsymbol{\mathcal{U}}$, $(\hat{s}_i, \hat{u}_i) \in \hat{\mathcal{S}}_i \times \hat{\mathcal{U}}_i$ ($i \in [1, k]$), and $\forall i, j \in [1, k], \hat{\mathcal{U}}_i \neq \hat{\mathcal{U}}_j$ if $i \neq j$. $\forall i \in [1, k]$, $\hat{\mathcal{S}}_i$ and $\hat{\mathcal{U}}_i$ denote a subset of the state space and a non-empty proper subset of the joint action space, respectively, i.e., $\hat{\mathcal{S}}_i \subset \mathcal{S}$, $\hat{\mathcal{U}}_i \subsetneqq \boldsymbol{\mathcal{U}}$ and $\hat{\mathcal{U}}_i \neq \emptyset$. Here we prove that $\mathcal{MG}$ is decomposable by $\{\mathcal{MG}_1, \mathcal{MG}_2, \cdots, \mathcal{MG}_k\}$, where $\mathcal{MG}_i := \ < \hat{\mathcal{S}}_i, \hat{\mathcal{U}}_i, \mathcal{P}, r_i, n_i, \gamma > (i \in [1, k])$.

*Proof.* Firstly, consider the action-value function under sarsa value iteration. We have

$$
\begin{aligned}
\mathcal{Q}(s_t, \boldsymbol{u}_t) &= \sum_{i=1}^{k} \mathcal{Q}_i(\hat{s}_{i,t}, \hat{u}_{i,t}) \\
&= r(s_t, \boldsymbol{u}_t) + \gamma \int_{s_{t+1}} \mathcal{P}(s_{t+1}|s_t, \boldsymbol{u}_t) V(s_{t+1}) ds_{t+1} \\
&= r(s_t, \boldsymbol{u}_t) + \gamma \int_{s_{t+1}} \mathcal{P}(s_{t+1}|s_t, \boldsymbol{u}_t) \int_{\boldsymbol{u}_{t+1}} \pi(\boldsymbol{u}_{t+1}|s_{t+1}) \mathcal{Q}(s_{t+1}, \boldsymbol{u}_{t+1}) d\boldsymbol{u}_{t+1} ds_{t+1} \\
&= r(s_t, \boldsymbol{u}_t) + \gamma \sum_{i=1}^{k} \int_{s_{t+1}} \mathcal{P}(s_{t+1}|s_t, \boldsymbol{u}_t) \int_{\boldsymbol{u}_{t+1}} \pi(\boldsymbol{u}_{t+1}|s_{t+1}) \mathcal{Q}_i(\hat{s}_{i,t+1}, \hat{u}_{i,t+1}) d\boldsymbol{u}_{t+1} ds_{t+1}
\end{aligned}
\tag{29}
$$

Since the environment is fully observable, we have $\mathcal{Q}_i(\hat{s}_{i,t+1}, \hat{u}_{i,t+1}) = \mathcal{Q}_i(s_{t+1}, \hat{u}_{i,t+1})$. Let $\hat{\pi}_i(\hat{u}_{i,t+1}|s_{t+1})$ denote the joint policy of the agents whose actions are involved in $\hat{u}_{i,t+1}$. Note $\hat{\pi}_i(\hat{u}_{i,t+1}|s_{t+1})$ is generally defined and determined by $\mathcal{Q}_i(s_{t+1}, \hat{u}_{i,t+1})$ for on-policy data distribution. Since $\mathcal{Q}_i(\hat{s}_{i,t+1}, \hat{u}_{i,t+1}) = \mathcal{Q}_i(s_{t+1}, \hat{u}_{i,t+1})$, we have $\hat{\pi}_i(\hat{u}_{i,t+1}|s_{t+1}) = \hat{\pi}_i(\hat{u}_{i,t+1}|\hat{s}_{i,t+1})$. Substituting this equality into Eq.22, we have

$$
\int_{\boldsymbol{u}_{t+1}} \pi(\boldsymbol{u}_{t+1}|s_{t+1}) \mathcal{Q}_i(\hat{s}_{i,t+1}, \hat{u}_{i,t+1}) d\boldsymbol{u}_{t+1} = \int_{\hat{u}_{i,t+1}} \hat{\pi}_i(\hat{u}_{i,t+1}|\hat{s}_{i,t+1}) \mathcal{Q}_i(\hat{s}_{i,t+1}, \hat{u}_{i,t+1}) d\hat{u}_{i,t+1}
\tag{30}
$$

Let $V_i(\hat{s}_{i,t+1})$ denote the right side of Eq.30. Substituting $V_i(\hat{s}_{i,t+1})$ into Eq.29, we have

$$
\begin{aligned}
\sum_{i=1}^{k} \mathcal{Q}_i(\hat{s}_{i,t}, \hat{u}_{i,t}) &= r(s_t, \boldsymbol{u}_t) + \gamma \sum_{i=1}^{k} \int_{s_{t+1}} \mathcal{P}(s_{t+1}|s_t, \boldsymbol{u}_t) V_i(\hat{s}_{i,t+1}) ds_{t+1} \\
&= r(s_t, \boldsymbol{u}_t) + \gamma \sum_{i=1}^{k} \int_{\hat{s}_{i,t+1}} \mathcal{P}(\hat{s}_{i,t+1}|s_t, \boldsymbol{u}_t) V_i(\hat{s}_{i,t+1}) d\hat{s}_{i,t+1}
\end{aligned}
\tag{31}
$$

Let $\mathcal{Q}'(s_t, \boldsymbol{u}_t) = \sum_{i=1}^{k} \mathcal{Q}'_i(\hat{s}_{i,t}, \hat{u}_{i,t})$ denote the action-value function with respect to policy $\pi'(\boldsymbol{u}_{t+1}|s_{t+1})$. We have

$$
\begin{aligned}
\mathcal{Q}'(s_t, \boldsymbol{u}_t) - \mathcal{Q}(s_t, \boldsymbol{u}_t) &= \sum_{i=1}^{k} [\mathcal{Q}'_i(\hat{s}_{i,t}, \hat{u}_{i,t}) - \mathcal{Q}_i(\hat{s}_{i,t}, \hat{u}_{i,t})] \\
&= \gamma \sum_{i=1}^{k} \int_{\hat{s}_{i,t+1}} \mathcal{P}(\hat{s}_{i,t+1}|s_t, \boldsymbol{u}_t) \left[ V'_i(\hat{s}_{i,t+1}) - V_i(\hat{s}_{i,t+1}) \right] d\hat{s}_{i,t+1}
\end{aligned}
\tag{32}
$$

where $V'_i(\hat{s}_{i,t+1}) = \int_{\hat{u}_{i,t+1}} \hat{\pi}'_i(\hat{u}_{i,t+1}|\hat{s}_{i,t+1}) \mathcal{Q}'_i(\hat{s}_{i,t+1}, \hat{u}_{i,t+1}) d\hat{u}_{i,t+1}$. A necessary condition of the equality above is $\forall p, q \in [1, k], \mathcal{P}(\hat{s}_{p,t+1}|s_t, \boldsymbol{u}_t) = \mathcal{P}(\hat{s}_{p,t+1}|\hat{s}_{q,t}, \hat{u}_{q,t})$.

Assume $\exists p, q \in [1, k]$ $(p \neq q)$, such that $\mathcal{P}(\hat{s}_{p,t+1}|s_t, \boldsymbol{u}_t) = \mathcal{P}(\hat{s}_{p,t+1}|\hat{s}_{q,t}, \hat{u}_{q,t})$. Note that $\hat{\mathcal{U}}_p \neq \hat{\mathcal{U}}_q$ since $p \neq q$. If we let $\hat{\pi}'_q(\hat{u}_{q,t+1}|\hat{s}_{q,t+1}) = \hat{\pi}_q(\hat{u}_{q,t+1}|\hat{s}_{q,t+1})$ while $\hat{\pi}'_p(\hat{u}_{p,t+1}|\hat{s}_{p,t+1}) \neq \hat{\pi}_p(\hat{u}_{p,t+1}|\hat{s}_{p,t+1})$, the term with $(\hat{s}_{q,t}, \hat{u}_{q,t})$ in the left side of Eq.32, i.e., $\mathcal{Q}'_q(\hat{s}_{q,t}, \hat{u}_{q,t}) - \mathcal{Q}_q(\hat{s}_{q,t}, \hat{u}_{q,t})$ is eliminated. However, the term with $(\hat{s}_{q,t}, \hat{u}_{q,t})$ in the right side of the equality, i.e., $\int_{\hat{s}_{p,t+1}} \mathcal{P}(\hat{s}_{p,t+1}|s_t, \boldsymbol{u}_t) [V'_i(\hat{s}_{p,t+1}) - V_i(\hat{s}_{p,t+1})] d\hat{s}_{p,t+1} = \int_{\hat{s}_{p,t+1}} \mathcal{P}(\hat{s}_{p,t+1}|\hat{s}_{q,t}, \hat{u}_{q,t}) [V'_i(\hat{s}_{p,t+1}) - V_i(\hat{s}_{p,t+1})] d\hat{s}_{p,t+1}$ remains. Therefore, Eq.32 holds only if $\forall p, q \in [1, k]$, $\mathcal{P}(\hat{s}_{p,t+1}|s_t, \boldsymbol{u}_t) = \mathcal{P}(\hat{s}_{p,t+1}|\hat{s}_{q,t}, \hat{u}_{q,t})$ and $p = q$, i.e.,

- $\forall i \in [1, k]$, $\mathcal{P}(\hat{s}_{i,t+1}|s_t, \boldsymbol{u}_t) = \mathcal{P}(\hat{s}_{i,t+1}|\hat{s}_{i,t}, \hat{u}_{i,t})$

Substituting the equality above into Eq.31, we have

$$r(s_t, \boldsymbol{u}_t) = \sum_{i=1}^{k} \left[ \mathcal{Q}_i(\hat{s}_{i,t}, \hat{u}_{i,t}) - \gamma \int_{\hat{s}_{i,t+1}} \mathcal{P}(\hat{s}_{i,t+1}|\hat{s}_{i,t}, \hat{u}_{i,t}) V_i(\hat{s}_{i,t+1}) d\hat{s}_{i,t+1} \right] \quad (33)$$

Let $r_i(\hat{s}_{i,t}, \hat{u}_{i,t}) := \mathcal{Q}_i(\hat{s}_{i,t}, \hat{u}_{i,t}) - \gamma \int_{\hat{s}_{i,t+1}} \mathcal{P}(\hat{s}_{i,t+1}|\hat{s}_{i,t}, \hat{u}_{i,t}) V_i(\hat{s}_{i,t+1}) d\hat{s}_{i,t+1}$. We have

- $r(s_t, \boldsymbol{u}_t) = \sum_{i=1}^{k} r_i(\hat{s}_{i,t}, \hat{u}_{i,t})$

According to Definition 3.1, $\mathcal{MG}$ is decomposable on $\{\hat{\mathcal{S}}_1 \times \hat{\mathcal{U}}_1, \cdots, \hat{\mathcal{S}}_k \times \hat{\mathcal{U}}_k\}$.

For action-value function under the Q-learning value iteration, we have

$$\begin{aligned}
\mathcal{Q}(s_t, \boldsymbol{u}_t) &= \sum_{i=1}^{k} \mathcal{Q}_i(\hat{s}_{i,t}, \hat{u}_{i,t}) \\
&= r(s_t, \boldsymbol{u}_t) + \gamma \int_{s_{t+1}} \mathcal{P}(s_{t+1}|s_t, \boldsymbol{u}_t) \max_{\hat{\boldsymbol{u}}_{t+1} \in \boldsymbol{\mathcal{U}}} \mathcal{Q}(s_{t+1}, \boldsymbol{u}_{t+1}) ds_{t+1} \\
&= r(s_t, \boldsymbol{u}_t) + \gamma \sum_{i=1}^{k} \int_{s_{t+1}} \mathcal{P}(s_{t+1}|s_t, \boldsymbol{u}_t) \max_{\hat{u}_{i,t+1} \in \hat{\mathcal{U}}_i} \mathcal{Q}_i(\hat{s}_{i,t+1}, \hat{u}_{i,t+1}) ds_{t+1} \\
&= r(s_t, \boldsymbol{u}_t) + \gamma \sum_{i=1}^{k} \int_{\hat{s}_{i,t+1}} \mathcal{P}(\hat{s}_{i,t+1}|s_t, \boldsymbol{u}_t) \max_{\hat{u}_{i,t+1} \in \hat{\mathcal{U}}_i} \mathcal{Q}_i(\hat{s}_{i,t+1}, \hat{u}_{i,t+1}) d\hat{s}_{i,t+1}
\end{aligned} \quad (34)$$

Replacing $V_i(\hat{s}_{i,t+1})$ with $\max_{\hat{u}_{i,t+1} \in \hat{\mathcal{U}}_i} \mathcal{Q}_i(\hat{s}_{i,t+1}, \hat{u}_{i,t+1})$ and following the deduction from Eq.31 to Eq.33, the decomposability of $\mathcal{MG}$ for Q-learning value iteration can be proved. Here we omit the details. $\qquad \square$

## D   Examples of Decomposable & Indecomposable MMDPs

## E   Representation for LMF in Indecomposable MMDP: Solving an Overdetermined Linear Equation System

Consider the discrete action space setting. At each state, the complete representation of action-values is equivalent to solving the following linear equation system

$$\left\{ \mathcal{Q}(s, \boldsymbol{u}) = \sum_{a=1}^{n} Q_a(s, u_a) \right\}_{\forall \boldsymbol{u} \in \boldsymbol{\mathcal{U}}} \quad (35)$$

Here we prove that the equation system is overdetermined.

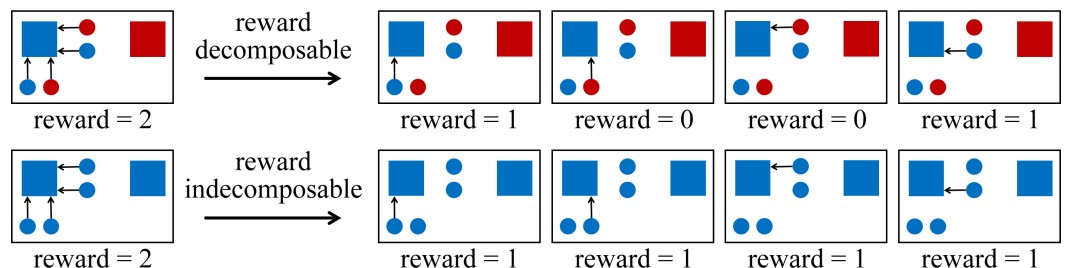

Figure 9: Examples of decomposable & indecomposable MMDPs. 4 agents (denoted by dots) need to cover 2 landmarks (denoted by squares). The agents would not collide with each other or any landmark. The team receives an instant reward when any agent arrives and gets accepted by a landmark. Each landmark only accepts the agent with the same color as it and is limited to accepting *no more than* 2 agents. The reward function is not linearly factorizable for the indecomposable case.

*Proof.* Let $\mathcal{U}_a := \{1, 2, \cdots, m\}$ ($\forall a \in [1, n]$) denote the discrete local action space. Each equation in Eq.35 can be represented by

$$
\begin{aligned}
\mathcal{Q}(s, \boldsymbol{u}) =& \mathbb{I}(u_1 = 1) \cdot Q_1(1) + \mathbb{I}(u_1 = 2) \cdot Q_1(2) + \cdots + \mathbb{I}(u_1 = m) \cdot Q_1(m) \\
&+ \mathbb{I}(u_2 = 1) \cdot Q_2(1) + \mathbb{I}(u_2 = 2) \cdot Q_2(2) + \cdots + \mathbb{I}(u_2 = m) \cdot Q_2(m) \\
&+ \cdots \\
&+ \mathbb{I}(u_n = 1) \cdot Q_n(1) + \mathbb{I}(u_n = 2) \cdot Q_n(2) + \cdots + \mathbb{I}(u_n = m) \cdot Q_n(m) \\
=& \left[ \overbrace{\mathbb{I}(u_1 = 1) \cdots \mathbb{I}(u_1 = m)}^{agent\ 1} \cdots \overbrace{\mathbb{I}(u_n = 1) \cdots \mathbb{I}(u_n = m)}^{agent\ n} \right] \\
&\times \left[ \overbrace{Q_1(1) \cdots Q_1(m)}^{agent\ 1} \cdots \overbrace{Q_n(1) \cdots Q_n(m)}^{agent\ n} \right]^{\top}
\end{aligned}
\tag{36}
$$

where the coefficient matrix $\left[ \overbrace{\mathbb{I}(u_1 = 1) \cdots \mathbb{I}(u_1 = m)}^{agent\ 1} \cdots \overbrace{\mathbb{I}(u_n = 1) \cdots \mathbb{I}(u_n = m)}^{agent\ n} \right]$ is determined by the permutation of local actions. Here we omit the states in the inputs of joint Q value functions. The equation system (Eq.35) is equivalent to a set of equations (Eq.36) under the all permutations of the joint action. For example, the all permutations of 2-agent local actions are

$$
\left[ \overbrace{(1, 1)\ (2, 1)\ \cdots\ (m, 1)}^{(\cdot, 1)}\ \overbrace{(1, 2)\ (2, 2)\ \cdots\ (m, 2)}^{(\cdot, 2)}\ \cdots\ \cdots\ \overbrace{(1, m)\ (2, m)\ \cdots\ (m, m)}^{(\cdot, m)} \right]^{\top}
\tag{37}
$$

By substituting Eq.37 into Eq.36, we obtain the matrix equation of Eq.35 in the two-agent case:

$$
\vec{\mathcal{Q}}^2 = \mathcal{A}^2 \times \vec{Q}_{loc}^2
\tag{38}
$$

where

$$
\mathcal{A}^2 = \begin{bmatrix} \overbrace{\begin{matrix} 1 & 0 & \cdots & 0 \\ 0 & 1 & \cdots & 0 \\ \vdots & \vdots & \ddots & \vdots \\ 0 & 0 & \cdots & 1 \\ & & \vdots & \\ 1 & 0 & \cdots & 0 \\ 0 & 1 & \cdots & 0 \\ \vdots & \vdots & \ddots & \vdots \\ 0 & 0 & \cdots & 1 \end{matrix}}^{agent\ 1} & \overbrace{\begin{matrix} 1 & 0 & \cdots & 0 \\ 1 & 0 & \cdots & 0 \\ \vdots & \vdots & \ddots & \vdots \\ 1 & 0 & \cdots & 0 \\ & \vdots & & \\ 0 & 0 & \cdots & 1 \\ 0 & 0 & \cdots & 1 \\ \vdots & \vdots & \ddots & \vdots \\ 0 & 0 & \cdots & 1 \end{matrix}}^{agent\ 2} \end{bmatrix}, \quad \vec{Q}_{loc}^2 = \begin{bmatrix} Q_1(1) \\ Q_1(2) \\ \vdots \\ Q_1(m) \\ Q_2(1) \\ Q_2(2) \\ \vdots \\ Q_2(m) \end{bmatrix}, \quad \vec{\mathcal{Q}}^2 = \begin{bmatrix} \mathcal{Q}(1, 1) \\ \mathcal{Q}(2, 1) \\ \vdots \\ \mathcal{Q}(m, 1) \\ \vdots \\ \vdots \\ \mathcal{Q}(1, m) \\ \mathcal{Q}(2, m) \\ \vdots \\ \mathcal{Q}(m, m) \end{bmatrix}
\tag{39}
$$

The coefficient matrix $\mathcal{A}^2$ can be represented by

$$\mathcal{A}^2 = \begin{bmatrix} E_m & A_1^2 \\ E_m & A_2^2 \\ \vdots & \vdots \\ E_m & A_m^2 \end{bmatrix}, \quad A_i^2 = \begin{bmatrix} O_{i-}^2 & \vec{I}^2 & O_{i+}^2 \end{bmatrix} \tag{40}$$

where $E_m$ is an $m$-dimensional unit matrix. $O_{i-}^2$ and $O_{i+}^2$ are zero matrices of shape $m \times i$ and $m \times (m-i-1)$ ($i \in [0, m-1]$), respectively. $\vec{I}^2$ is a column vector of all 1 elements, whose length is $m$. Note that $rk\left(\begin{bmatrix} E_m & A_1^2 \\ E_m & A_2^2 \end{bmatrix}\right) = m + 1$. We have $rk(\mathcal{A}^2) = m + (m-1) = 2m - 1$. Now we extend the 2-agent case to the 3-agent, where

$$\mathcal{A}^3 = \begin{bmatrix} \mathcal{A}^2 & A_1^3 \\ \mathcal{A}^2 & A_2^3 \\ \vdots & \vdots \\ \mathcal{A}^2 & A_m^3 \end{bmatrix}, \quad A_i^3 = \begin{bmatrix} O_{i-}^3 & \vec{I}^3 & O_{i+}^3 \end{bmatrix} \tag{41}$$

$O_{i-}^3$ and $O_{i+}^3$ are zero matrices of shape $m^2 \times i$ and $m^2 \times (m-i-1)$ ($i \in [0, m-1]$), respectively. $\vec{I}^3$ is a column vector of all 1 elements, whose size is $m^2$. We have $rk(\mathcal{A}^3) = rk(\mathcal{A}^2) + m - 1 = 3m - 2$. For the $n$-agent case, we can infer that

$$rk(\mathcal{A}^n) = rk(\mathcal{A}^{n-1}) + m - 1 = rk(\mathcal{A}^2) + (n-2) \cdot (m-1) = n(m-1) + 1 \tag{42}$$

For an indecomposable MMDP, the maximum number of independent equations is $m^n$. Note that $m^n > n(m-1) + 1 \; \forall m, n \geq 2$. Therefore, the equation system is overdetermined. $\qquad\square$

# F  LMF IN INDECOMPOSABLE MMDP: UNBIASED TD TARGET UNDER SARSA VALUE ITERATION

*Proof.* At each state, the representation of the action-values by the joint Q value function of LMF is equivalent to solving a linear equation system as Eq.35. Referring to Eq.36, such an equation system can be represented by the following matrix equation in the $n$-agent case:

$$\vec{\mathcal{Q}}^n = \mathcal{A}^n \times \vec{Q}_{loc}^n \tag{43}$$

The expressions of $\mathcal{A}^n$, $\vec{Q}_{loc}^n$ and $\vec{\mathcal{Q}}^n$ can be inferred from Eq.35 and Eq.41. We consider the worst case where the augment matrix is full rank, i.e., $rk(\begin{bmatrix} \mathcal{A}^n & \vec{\mathcal{Q}}^n \end{bmatrix}) = m^n$. Note that $\forall m, n \geq 2$, $m^n > n(m-1) + 1$. The equation system is overdetermined, which can be solved by least square method. Let $\vec{\pi}^n$ denote the vector of the probabilities of the joint action's all permutations. We have $\sqrt{\vec{\pi}^n} \cdot (\mathcal{A}^n \times \vec{Q}_{loc}^n) = (\sqrt{\vec{\pi}^n} \cdot \mathcal{A}^n) \times \vec{Q}_{loc}^n$. The aim of the least square method is

$$min \;\; \vec{\pi}^n \cdot ||\mathcal{A}^n \times \vec{Q}_{loc}^n - \vec{\mathcal{Q}}^n|| = min \;\; ||(\sqrt{\vec{\pi}^n} \cdot \mathcal{A}^n) \times \vec{Q}_{loc}^n - \sqrt{\vec{\pi}^n} \cdot \vec{\mathcal{Q}}^n|| \tag{44}$$

According to the properties of the least square solution, $\vec{Q}_{loc}^{n*}$ is the least square solution if and only if

$$(\sqrt{\vec{\pi}^n} \cdot \mathcal{A}^n)^\top \times (\sqrt{\vec{\pi}^n} \cdot \mathcal{A}^n) \times \vec{Q}_{loc}^{n*} = (\sqrt{\vec{\pi}^n} \cdot \mathcal{A}^n)^\top \times (\sqrt{\vec{\pi}^n} \cdot \vec{\mathcal{Q}}^n) \tag{45}$$

Let $\vec{Q}_{tot}^{n*}$ denote the vector of the joint Q values of the joint action's all permutations under the least square solution. We have $\vec{Q}_{tot}^{n*} = \mathcal{A}^n \times \vec{Q}_{loc}^{n*}$. Therefore,

$$(\sqrt{\vec{\pi}^n} \cdot \mathcal{A}^n)^\top \times (\sqrt{\vec{\pi}^n} \cdot \mathcal{A}^n) \times \vec{Q}_{loc}^{n*} = (\sqrt{\vec{\pi}^n} \cdot \mathcal{A}^n)^\top \times (\sqrt{\vec{\pi}^n} \cdot \vec{Q}_{tot}^n) \tag{46}$$

Note the left sides of Eq.45 and Eq.46 are the same. We have

$$\begin{aligned}
(\sqrt{\vec{\pi}^n} \cdot \mathcal{A}^n)^\top \times (\sqrt{\vec{\pi}^n} \cdot \vec{\mathcal{Q}}^n) &= \mathcal{A}^{n\top} \times (\vec{\pi}^n \cdot \vec{\mathcal{Q}}^n) \\
= (\sqrt{\vec{\pi}^n} \cdot \mathcal{A}^n)^\top \times (\sqrt{\vec{\pi}^n} \cdot \vec{Q}_{tot}^n) &= \mathcal{A}^{n\top} \times (\vec{\pi}^n \cdot \vec{Q}_{tot}^n)
\end{aligned} \tag{47}$$

Referring to Eq.41, the coefficient matrix of the $n$-agent case equals

$$\mathcal{A}^{n\top} = \begin{bmatrix} \mathcal{A}^{n-1\top} & \mathcal{A}^{n-1\top} & \cdots & \mathcal{A}^{n-1\top} \\ A_1^{n\top} & A_2^{n\top} & \cdots & A_m^{n\top} \end{bmatrix}, \;\; A_i^{n\top} = \begin{bmatrix} O_{i-}^{n\top} \\ \vec{I}^{n\top} \\ O_{i+}^{n\top} \end{bmatrix} \tag{48}$$

where $O_{i-}^{n\top}$ and $O_{i+}^{n\top}$ are zero matrices of shape $i \times m^{n-1}$ and $(m-i-1) \times m^{n-1}$ ($i \in [0, m-1]$), respectively. $\vec{I}^{n\top}$ is a row vector of all 1 elements, whose length is $m^{n-1}$. Substituting Eq.48 into Eq.47, we have

$$\begin{bmatrix} A_1^{n\top} & A_2^{n\top} & \cdots & A_m^{n\top} \end{bmatrix} \times (\vec{\pi}^n \cdot \vec{Q}_{tot}^n) = \begin{bmatrix} A_1^{n\top} & A_2^{n\top} & \cdots & A_m^{n\top} \end{bmatrix} \times (\vec{\pi}^n \cdot \vec{\mathcal{Q}}^n) \tag{49}$$

Omit the zero elements in $A_i^{n\top}$ ($i \in [1, m]$). We have $\forall i \in [1, m]$,

$$\sum_{u_{\backslash 1}}^{\mathcal{U}_{\backslash 1}} \pi(u_1 = i, u_{\backslash 1}|s) Q(s, u_1 = i, u_{\backslash 1}) = \sum_{u_{\backslash 1}}^{\mathcal{U}_{\backslash 1}} \pi(u_1 = i, u_{\backslash 1}|s) \mathcal{Q}(s, u_1 = i, u_{\backslash 1}) \tag{50}$$

where $u_{\backslash 1}$ denotes the set of all actions except $u_1$, i.e., $u_{\backslash 1} \cup u_1 = \boldsymbol{u}$. Summing up the equations from $u_1 = 1$ to $u_1 = m$, we have

$$\sum_{i=1}^{m} \sum_{u_{\backslash 1}}^{\mathcal{U}_{\backslash 1}} \pi(u_1 = i, u_{\backslash 1}|s) Q(s, u_1 = i, u_{\backslash 1}) = \sum_{\boldsymbol{u}}^{\mathcal{U}} \pi(\boldsymbol{u}|s) Q(s, \boldsymbol{u})$$
$$= \sum_{i=1}^{m} \sum_{u_{\backslash 1}}^{\mathcal{U}_{\backslash 1}} \pi(u_1 = i, u_{\backslash 1}|s) \mathcal{Q}(s, u_1 = i, u_{\backslash 1}) = \sum_{\boldsymbol{u}}^{\mathcal{U}} \pi(\boldsymbol{u}|s) \mathcal{Q}(s, \boldsymbol{u}) \tag{51}$$

Let $V(s) = \sum_{\boldsymbol{u}}^{\mathcal{U}} \pi(\boldsymbol{u}|s) \mathcal{Q}(s, \boldsymbol{u})$ denote the actual state value function and $\widetilde{V}(s) = \sum_{\boldsymbol{u}}^{\mathcal{U}} \pi(\boldsymbol{u}|s) Q(s, \boldsymbol{u})$ denote the expectation of the joint Q value function obtained by LMF. According to Eq.51, we have $V(s) = \widetilde{V}(s)$. Temporal-Difference (TD) learning target of the joint Q value function under sarsa value iteration (i.e., sarsa target) is

$$y^{sarsa}(s_t, \boldsymbol{u}_t) = r_t + \gamma \sum_{s_{t+1}}^{\mathcal{S}} \mathcal{P}(s_{t+1}|s_t, \boldsymbol{u}_t) \cdot \widetilde{V}(s_{t+1})$$
$$= r_t + \gamma \sum_{s_{t+1}}^{\mathcal{S}} \mathcal{P}(s_{t+1}|s_t, \boldsymbol{u}_t) \cdot V(s_{t+1}) = \mathcal{Q}(s_t, \boldsymbol{u}_t) \tag{52}$$

Therefore, the sarsa target is an unbiased estimation of the action-value. $\qquad\square$

## G  RELATED WORKS

### G.1  DECOMPOSITION OF MULTI-AGENT GAMES

A previous work (Dou et al., 2022) also discusses the limitation of Linear Mixing Function (LMF) from the perspective of task properties and gives a definition to the decomposability of Multi-Agent Markov Game (MAMG). For convenience, let this work be referred to as W1. In W1, a *decomposable game* is defined as a MAMG with linearly factorizable reward and transition functions. The difference between W2 and our work lies in:

1. A decomposable game defined by W1 can only be decomposed agent by agent. By contrast, in our definition of decomposable MMDP, the task is decomposable by independent agent groups. Each group involves more than one agent.

2. The definition of the decomposable game defined by W1 involves the linear factorizability of the transition function, which is not intuitively explainable. The transition function describes the distribution of the next state conditioned on current state-action pair, which can not be linearly factorized. By contrast, our defined decomposability of MMDP is intuitively explicable. By decomposing the state-action space and reward function, the task is

decomposed into multiple independent sub-tasks. As a result, the action-value function of the whole task equals the sum of the action-value functions of all sub-tasks (as shown in Fig.2).

3. W1 only proves that LMF is free from representational limitation in decomposable games, but does not prove in what task the representational limitation occurs for LMF. By contrast, this paper prove a *sufficient and necessary* condition between the decomposability of an MMDP and the linear factorizability of the action-value function. Given the reward and transition functions, our definition of decomposable MMDP can be applied to distinguish whether the representational limitation occurs for LMF.

Besides, another work (Castellini et al., 2021) empirically investigate the learning power of various network architectures on a series of one-shot games. Let this work be referred to as W2. The difference between W2 and our work lies in:

1. W2 only carries out empirical evaluations to the learning power of different MARL methods, while our work carries out both theoretical analysis and empirical evaluations to the expressiveness of LMF.

2. W2 does not investigate learning strategies for repeated play, but only for the one-shot game. By contrast, our investigation is on the background of value iteration, which is a more complex case.

3. The investigated games in W2 are classified into factored and non-factored games, which are different to the decomposable and indecomposable MMDPs in our work. Firstly, the factored game in W2 is defined as the game with factorizable reward under specific conditions. But the decomposable MMDP in our work requires linearly factorizable reward and self-contained transitions.

## G.2 VALUE DECOMPOSITION

An early solution to fully cooperative multi-agent tasks is independent learning, e.g., Independent Q Learning (IQL) (Tan, 1993a). In tasks with a small number of agents, independent learning with agent-specific reward functions is able to acquire strategies on the level of human experts (de Witt et al., 2020; Berner et al., 2019). For better scalability, recent works turn to automatic credit assignment methods, e.g., value decomposition. Here we introduce value decomposition methods from the perspective of mixing functions. Note that the notations could be different from the original papers.

**Linear Mixing Function (LMF)**. LMF is firstly applied by VDN (Sunehag et al., 2017). The joint Q value function of VDN $Q_{tot}(\boldsymbol{\tau}, \boldsymbol{u}) = \sum_{a=1}^{n} Q_a(\tau_a, u_a)$ is trained to approximate the action-value function by Q-learning value iteration. QTRAN (Son et al., 2019) applies the LMF in another way. Specifically, a central Q value function $Q_{ct}(\boldsymbol{\tau}, \boldsymbol{u})$ (with complete representational capability) is introduced to approximate the action-value function by Q-learning value iteration. The difference between $Q_{ct}(\boldsymbol{\tau}, \boldsymbol{u})$ and $Q_{tot}(\boldsymbol{\tau}, \boldsymbol{u}_{gre})$ is denoted by $V_{ct}(\boldsymbol{\tau})$, i.e., $V_{ct}(\boldsymbol{\tau}) := Q_{ct}(\boldsymbol{\tau}, \boldsymbol{u}_{gre}) - Q_{tot}(\boldsymbol{\tau}, \boldsymbol{u}_{gre})$. $Q_{tot}(\boldsymbol{\tau}, \boldsymbol{u})$ is trained with the target $y := Q_{ct}(\boldsymbol{\tau}, \boldsymbol{u}) - V_{ct}(\boldsymbol{\tau})$ and would not updated if $Q_{tot}(\boldsymbol{\tau}, \boldsymbol{u}) > y$. QTRAN introduces the target $y$ to distinguish whether $\mathcal{Q}(s, \boldsymbol{u})$ is underestimated by $Q_{tot}(\boldsymbol{\tau}, \boldsymbol{u})$. $Q_{tot}(\boldsymbol{\tau}, \boldsymbol{u})$ is updated only in underestimated cases. Since the optimal action-value is more likely to be underestimated, QTRAN alleviates the optimal representation interference (ORI) to some extent.

**Strictly Monotonic Mixing Function (SMMF)**. QMIX (Rashid et al., 2020b) introduces SMMF in value decomposition. The joint Q value function of QMIX is trained to approximate the action-value function by Q-learning value iteration. SMIX Yao et al. (2021), Qatten Yang et al. (2020) and WQMIX (Rashid et al., 2020a) apply the same mixing function with QMIX. SMIX replaces the TD(0) Q-learning target with a TD($\lambda$) sarsa target. Qatten introduces an attention network before the mixing network. WQMIX (Rashid et al., 2020a) implements the mixing function of QMIX in a QTRAN-like manner. Specifically, WQMIX introduces a central Q value function $Q_{ct}(s, \boldsymbol{\tau}, \boldsymbol{u})$ (with complete representational capability) to approximate the action-value function by Q-learning value iteration. $Q_{ct}(s, \boldsymbol{\tau}, \boldsymbol{u})$ and the joint Q value function $Q_{tot}(s, \boldsymbol{\tau}, \boldsymbol{u})$ are trained with the same target $y = r(s, \boldsymbol{u}) + \gamma Q_{ct}(s', \boldsymbol{\tau}', \boldsymbol{u}'_{gre})$. WQMIX introduces a small weight to the loss function if $Q_{ct}(s, \boldsymbol{\tau}, \boldsymbol{u}_{gre}) > y$ (CW-QMIX) or $Q_{tot}(ss, \boldsymbol{\tau}, \boldsymbol{u}) > y$ (OW-QMIX). In words, WQMIX places

more importance on the representation of underestimated action-values, which also alleviates the ORI to some extent. Both QTRAN and WQMIX rely on the accurate discrimination of underestimated action-values.

**Other Mixing Functions**. DCG (Böhmer et al., 2020) introduces the concept of a coordination graph, which factorizes the joint Q value function by nodes (local Q value functions) and edges (Q value function conditioned on agent groups). The mixing function of DCG has complete representational capability only if the graph is fully connected. Otherwise, DCG suffers from representational limitation in indecomposable MMDPs, where the action-value functions are not linearly factorizable by any group of agents. However, a fully connected graph brings high computational cost. QPLEX (Wang et al., 2020b) propose a dueling mixing network. As discussed in Section 4.1, the mixing function of QPLEX can be viewed as an implementation of the difference mixing framework. The mixing function of QPLEX has complete representational capability but would suffer from severe Optimal Representational Interference (ORI). Detailed discussion of the ORI of QPLEX mixing function can be found in Appendix I.

A summary of value decomposition methods is provided in Tab.1.

Table 1: A summary of value decomposition methods. LMF: Linear Mixing Function. SMMF: Strictly Monotonic Mixing Function. MUD: Mixing for Unbounded Difference. ORI: Optimal Representation Interference. FC: Fully Connected.

| Method | Mixing Function | Monotonic | IGM | Complete Expressiveness | Addressing ORI |
|---|---|---|---|---|---|
| VDN | LMF | √ | √ | × | × |
| QTRAN | LMF | √ | √ | × | √ |
| QMIX | SMMF | √ | √ | × | × |
| SMIX | SMMF | √ | √ | × | × |
| Qatten | SMMF | √ | √ | × | × |
| WQMIX | SMMF | √ | √ | × | √ |
| DCG | N/A | √ | √ | FC graph | × |
| QPLEX | MUD | √ | √ | √ | × |
| MUD-SmG (ours) | MUD | √ | √ | √ | √ |
| MUD-StG (ours) | MUD | √ | √ | √ | √ |

# H  NON-MONOTONIC MIXING FUNCTIONS SUFFERS FROM POOR CONVERGENCE

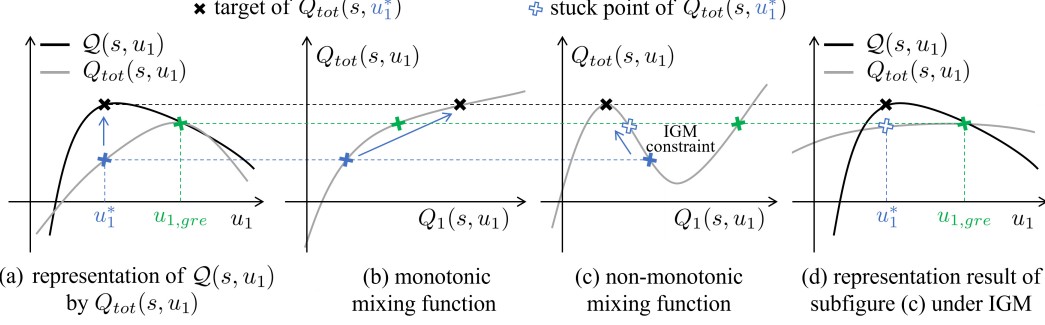

(a) representation of $\mathcal{Q}(s, u_1)$ by $Q_{tot}(s, u_1)$

(b) monotonic mixing function

(c) non-monotonic mixing function

(d) representation result of subfigure (c) under IGM

Figure 10: The joint Q value function is easily stuck in sub-optimums for non-monotonic mixing functions.

Consider the value decomposition in fully cooperative MARL problem. Suppose all agents except agent 1 take their greedy actions, which we omit in the notation. Let $u_1^* = \arg\max_{u_1} \mathcal{Q}(s, u_1)$ and $u_{1,gre} = \arg\max_{u_1} Q_{tot}(s, u_1)$ denote the optimal and greedy actions, respectively. As shown in Fig.10:

(a) $Q_{tot}(s, u_1^*)$ (blue cross) is trained to represent its target $\mathcal{Q}(s, u_1^*)$ (black cross)

(b) For the monotonic mixing function, $Q_{tot}(s, u_1^*)$ is able to reach the target under the IGM constraint.

(c) For the non-monotonic mixing function, we have $\arg\max_{u_1} Q_{tot}(s, u_1) = u_1^*$ but $\arg\max_{u_1} Q_1(s, u_1) = u_{1,gre}$ if $Q_{tot}(s, u_1^*)$ reaches the target, for which the IGM (Eq.2) would be violated.

(d) Since the IGM constraint holds, $Q_{tot}(s, u_1^*)$ could not reach the target, but would be stuck in the sub-optimum (hollow blue cross) for the non-monotonic mixing function.

## I  EXAMPLE OF THE ORI IN MUD: QPLEX MIXING FUNCTION (WITHOUT STOPPED GRADIENT)

An example of ORI is the mixing function of QPLEX (Wang et al., 2020b) without stopped gradient. Consider the fully observable setup. The joint Q value function of QPLEX is

$$Q_{tot}(s, \boldsymbol{u}) = \sum_{a=1}^{n} Q_a(s, u_{a,gre}) - \sum_{a=1}^{n} w_a(s, \boldsymbol{u}) \cdot [Q_a(s, u_{a,gre}) - Q_a(s, u_a)] \tag{53}$$

where $w_a(s, \boldsymbol{u}) \geq 0$. Note that $\mathcal{F}'_a(s, \boldsymbol{u}) = w_a(s, \boldsymbol{u})$ and $\frac{Q_{tot}(s,\boldsymbol{u})}{w_a(s,\boldsymbol{u})} = -[Q_a(s, u_{a,gre}) - Q_a(s, u_a)] < 0$. Therefore, $\mathcal{F}'_a(s, \boldsymbol{u})$ decreases as $Q_{tot}(s, \boldsymbol{u})$. As a result, the optimal representation ratio $\boldsymbol{w}^*$ could be extremely small, which leads to sever ORI.

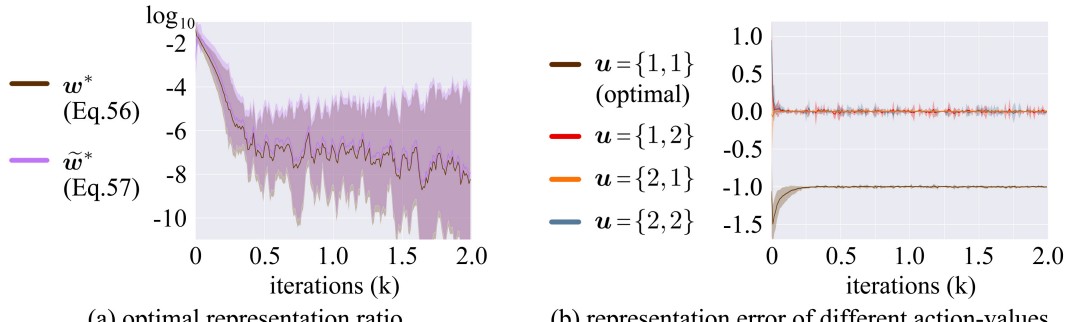

(a) optimal representation ratio  (b) representation error of different action-values

Figure 11: Failure cases of QPLEX mixing function (without stopped gradient) on a 2-agent matrix game. (a) the optimal representation ratio $\boldsymbol{w}^*$ decreases to an extremely low level (around 10e-8). (b) as a result, the representational error of the optimal action-value does not decrease.

Here we introduce an example to illustrate the problem. Consider a two-agent matrix game with the payoff matrix (i.e., action-value matrix) $[[1, -1], [-1, 0]]$. Assume the initial greedy action is $\boldsymbol{u}_{gre} = \{2, 2\}$, where $u_1, u_2 \in \{1, 2\}$. The representation of the matrix is equivalent to solving the following equation system:

$$\begin{cases} -w_1(1,1) \cdot [Q_1(2) - Q_1(1)] - w_2(1,1) \cdot [Q_2(2) - Q_2(1)] + Q_1(2) + Q_2(2) = 1 \\ -w_1(1,2) \cdot [Q_1(2) - Q_1(1)] \qquad\qquad\qquad\qquad\qquad\quad + Q_1(2) + Q_2(2) = -1 \\ \qquad\qquad\qquad\qquad -w_2(2,1) \cdot [Q_2(2) - Q_2(1)] + Q_1(2) + Q_2(2) = -1 \\ \qquad\qquad\qquad\qquad\qquad\qquad\qquad\qquad\qquad\qquad\quad Q_1(2) + Q_2(2) = 0 \end{cases} \tag{54}$$

where we omit the input states. The first equality is unachievable without the change of the greedy action. Assuming the other 3 equalities hold and the action distribution is uniform, the optimal representation ratio $\boldsymbol{w}^*$ equals

$$\boldsymbol{w}^* = \frac{1}{2} \left[ \frac{w_1(1,1)}{w_1(1,1) + w_1(1,2)} + \frac{w_2(1,1)}{w_2(1,1) + w_2(2,1)} \right] \tag{55}$$

which can be approximated by

$$\widetilde{\boldsymbol{w}}^* = \frac{1}{2} \left[ \frac{w_1(1,1) \cdot [Q_1(2) - Q_1(1)]}{w_1(1,1) \cdot [Q_1(2) - Q_1(1)] + 1} + \frac{w_2(1,1) \cdot [Q_2(2) - Q_2(1)]}{w_2(1,1) \cdot [Q_2(2) - Q_2(1)] + 1} \right] \approx \boldsymbol{w}^* \tag{56}$$

Such approximation is evaluated in Fig.11(b). To approximate the first equality in Eq.54, $\forall a \in \{1, 2\}$, $w_a(1, 1) \cdot [Q_a(2) - Q_a(1)]$ would decrease to 0, which leads to an extremely low $\boldsymbol{w}^*$. As a result, the representation of $\mathcal{Q}(1, 1)$ is overwhelmed by the representation of $\mathcal{Q}(1, 2)$ and $\mathcal{Q}(2, 1)$.

We carry out experiments to verify the ORI of QPLEX mixing function (without stopped gradient). The joint Q value function is implemented by neural networks. Each module is implemented by a 2-layer MLP. $Q_{tot}(s, \boldsymbol{u})$ is trained by Q-learning value iteration in an on-policy manner. We adopt the $\epsilon$-greedy strategy to sample the joint action and set $\epsilon = 0.2$ throughout the training. In 50 times of independent training, the greedy action gets trapped in $\boldsymbol{u}_{gre} = \{1, 1\}$ for 24 times. We visualize the training process of the 24 failure cases, which is shown in Fig.11. Actually, to avoid $\boldsymbol{w}^*$ decreasing, QPLEX applies the following mixing function

$$Q_{tot}(s, \boldsymbol{u}) = \sum_{a=1}^{n} Q_a(s, u_a) - \sum_{a=1}^{n} [w_a(s, \boldsymbol{u}) - 1] \cdot [Q_a(s, u_{a, gre}) - SG(Q_a(s, u_a))] \qquad (57)$$

where $SG$ refers to stopping gradient. For Eq.57, we have $\mathcal{F}'_a(s, \boldsymbol{u}) = \frac{Q_{tot}(s, \boldsymbol{u})}{Q_a(s, u_a)} = 1$. According to Eq.11, $\boldsymbol{w}^*$ depends only on the action distribution, (i.e., policy $\pi(\boldsymbol{u}|s)$).

## J EXPERIMENTAL SETUPS

### J.1 VERIFICATION OF THE EXPRESSIVENESS OF LMF

**Detailed Task setup**. Referring to Fig.2 and the examples in Fig.9, we design toy games for both decomposable and indecomposable MMDPs. 4 agents (denoted by dots) need to cover 2 landmarks (denoted by squares) in pairs. The map is gridded by a $4 \times 4$ checkerboard. All agents are initialized with the position $(3, 0)$ and required to select actions from $\{up, right\}$ at each time step. We mask invalid actions, e.g., $up$ at position $(0, 0)$. The task is fully observable and the state consists of the positions of all agents. The agents would not collide with each other or any landmark. The team receives an instant reward of 1 when any agent arrives and gets accepted by a landmark. Each landmark only accepts the agent with the same color as it and is limited to accept *no more than* 2 agents. As explained in Fig.9, the reward function is not linearly factorizable for the indecomposable case.

**Training setup**. We introduce a central Q value function $Q_{ct}(s, \boldsymbol{u})$ modelled by a 5-layer neural network to learn the action-value function. Besides, we also apply neural networks to model two joint Q value functions: (1) $Q_{lmf}(s, \boldsymbol{u}) = \sum_{a=1}^{4} Q_a(s, u_a)$; (2) $Q_{33} = Q'_1(s, u_1, u_2, u_3) + Q'_2(s, u_1, u_2, u_4)$. $Q_{ct}(s, \boldsymbol{u})$, $Q_{lmf}(s, \boldsymbol{u})$ and $Q_{33}$ are trained by both sarsa and Q-learning value iteration. We test the Root Mean Square Errors (RMSE) of $Q_{lmf}(s, \boldsymbol{u})$ and $Q_{33}(s, \boldsymbol{u})$ to $\mathcal{Q}(s, \boldsymbol{u})$. Formally,

$$RMSE_{33}(s) = \left( \frac{1}{m^n} \sum_{\boldsymbol{u}}^{\boldsymbol{\mathcal{U}}} [Q_{ct}(s, \boldsymbol{u}) - Q_{33}(s, \boldsymbol{u}))]^2 \right)^{\frac{1}{2}} \qquad (58)$$

Note we approximate $\mathcal{Q}(s, \boldsymbol{u})$ with a central joint Q value function $Q_{ct}(s, \boldsymbol{u})$. $RMSE_{lmf}(s)$ can be obtained by replacing $Q_{33}(s, \boldsymbol{u})$ with $Q_{lmf}(s, \boldsymbol{u})$ in Eq.58. The RMSE is compared with the test mean action-value $V_{ct}(s)$, which is defined as $V_{ct}(s) := \frac{1}{m^n} \sum_{\boldsymbol{u}}^{\boldsymbol{\mathcal{U}}^n} Q_{ct}(s, \boldsymbol{u})$. For the experiment of Fig.6(b), we adopt the $\epsilon$-greedy exploration strategy and set $\epsilon = 0.2$ through out the training. For the experiment of Fig.6(c), all agents follow random policies. For the experiments of Fig.12, actions are sampled from a predefined action distribution, which is randomly generated at the beginning of training.

### J.2 SINGLE-STEP MATRIX GAMES

All modules of the joint Q value functions are implemented by 2-layer neural networks. We adopt the $\epsilon$-greedy exploration strategy and set $\epsilon = 0.2$ through out the training. All joint Q value functions are trained in an on-policy manner. The expressions of the mixing functions are provided in Tab.2. $\alpha = 1.0$ for MUD-StG. For better stability, we implement the central bias $b(s, \boldsymbol{u})$ in Eq.12 and Eq.13 by 2-layer MLPs directly and constrain that $b(s, \boldsymbol{u}) > 0$ by an absolute value function. Note that consequently, the IGM would be violated. To approximate the IGM, the regularization $b(s, \boldsymbol{u}_{gre}) = 0$ is applied to the loss function.

Table 2: The expressions of the mixing functions. For brevity, we denote $Q(s, u_a)$ and $Q(s, u_{a,gre})$ by $Q_a$ and $Q_{a,gre}$, respectively. $\vec{Q}_{locs} = [Q_1, \cdots, Q_n]$. $\times$ denotes matrix product. $f_{ac}$ is a monotonic activation function. $b(s, \boldsymbol{u}) = |\vec{h} - \vec{h}_{gre}|_2$, where $\vec{h}$ and $\vec{h}_{gre}$ are vectors generated from $(s, \boldsymbol{u})$ and $(s, \boldsymbol{u}_{gre})$, respectively. $SG$ refers to stopped gradient. $\alpha$ is a hyper-parameter. The following variables are constraint to be absolute values: (1) elements of $\vec{w}_1(s)$ and $\vec{w}_2(s)$ (SMMF); (2) $w(s, \boldsymbol{u})$ (MUD); (3) $w_a(s, \boldsymbol{u})$ (MUD (QPLEX) and MUD (QPLEX-SG)); (4) $Q_a$ and $Q_{a,gre}$ (MUD-SmG). The shapes of the variables of SMMF is provided in Tab.3.

| Mixing Function | Expressions of $Q_{tot}(\boldsymbol{s}, \boldsymbol{u})$ |
|---|---|
| LMF | $\sum_{a=1}^n Q_a$ |
| SMMF | $\vec{w}_2(s) \times f_{ac}\left(\vec{Q}_{locs} \times \vec{w}_1(s) + \vec{b}_1(s)\right)^{\top} + b_2(s)$ |
| MUD (Eq.9) | $V(s) - \sum_{a=1}^n w(s, \boldsymbol{u}) \cdot (Q_{a,gre} - Q_a) + b(s, \boldsymbol{u})$ |
| MUD (QPLEX) | $\sum_{a=1}^n Q_{a,gre} - \sum_{a=1}^n w_a(s, \boldsymbol{u}) \cdot (Q_{a,gre} - Q_a)$ |
| MUD (QPLEX-SG) | $\sum_{a=1}^n Q_a - \sum_{a=1}^n [w_a(s, \boldsymbol{u}) - 1] \cdot (Q_{a,gre} - SG(Q_a))$ |
| MUD-SmG | $\sum_{a=1}^n Q_{a,gre} - e^{-b(s,\boldsymbol{u})} \cdot \frac{1}{n} \sum_{a=1}^n \left(1 - \frac{Q_a}{Q_{a,gre}}\right) - b(s, \boldsymbol{u})$ |
| MUD-StG | $\begin{array}{ll} \sum_{a=1}^n (Q_{a,gre} - SG(Q_a)) + b(s, \boldsymbol{u}) & b(s, \boldsymbol{u}) > \alpha \\ \sum_{a=1}^n (Q_{a,gre} - Q_a) + b(s, \boldsymbol{u}) & otherwise \end{array}$ |

Table 3: Shapes of the variables of SMMF. $h \in \{1, 2, 3, \cdots\}$.

| variable | $\vec{Q}_{locs}$ | $\vec{w}_1(s)$ | $b_1(s)$ | $\vec{w}_2(s)$ | $b_2(s)$ |
|---|---|---|---|---|---|
| shape | $(1, n)$ | $(n, h)$ | $(1, h)$ | $(1, h)$ | $(1, 1)$ |

### J.3 PREDATOR PREY AND SMAC

We adopt the same task and training setups as WQMIX (Rashid et al., 2020a) in all experiments.

## K ADDITIONAL EXPERIMENTS

### K.1 ESTIMATION BIAS OF THE STATE VALUE FOR LMF IN INDECOMPOSABLE MMDP

We apply our designed toy game (Fig.5) to verify our finding in Appendix F, i.e., the sarsa target is an unbiased estimation of the action-value for LMF in indecomposable MMDP. To be specific, we test the difference between $V_{ct}(s) = \sum_{\boldsymbol{u}}^{\boldsymbol{\mathcal{U}}} \pi(\boldsymbol{u}|s) Q_{ct}(s, \boldsymbol{u})$ and $\widetilde{V}_{lmf}(s) = \sum_{\boldsymbol{u}}^{\boldsymbol{\mathcal{U}}} \pi(\boldsymbol{u}|s) Q_{lmf}(s, \boldsymbol{u})$ under sarsa value iteration in the indecomposable case. $Q_{lmf}(s, \boldsymbol{u}) = \sum_{a=1}^n Q_a(s, u_a)$. $Q_{ct}(s, \boldsymbol{u})$ is a central joint Q value function applied to approximate the action-value function. We also test the estimation error of Q-learning targets, i.e., $\Delta Q_{max,lmf}^Q(s) = max_{\boldsymbol{u} \in \boldsymbol{\mathcal{U}}} Q_{lmf}(s, \boldsymbol{u}) - max_{\boldsymbol{u} \in \boldsymbol{\mathcal{U}}} Q_{ct}(s, \boldsymbol{u})$ under Q-learning value iteration. Only the 98 states of first 3 time steps are recorded. We visualize the test results after 6k iterations of training, which are shown in Fig.12. Each bar denotes the test result of a single state. Detailed task and training setups are available in Appendix J.1.

### K.2 STARCRAFT MULTI-AGENT CHALLENGE

we compare MUD-SmG and MUD-StG with other value decomposition methods on challenging tasks of the StarCraft Multi-Agent Challenge (SMAC) (Samvelyan et al., 2019). We set $\alpha = 1.0$ for MUD-StG. The experimental results are shown in Fig.13. Fig.13 demonstrates that MUD-StG outperforms the other methods.

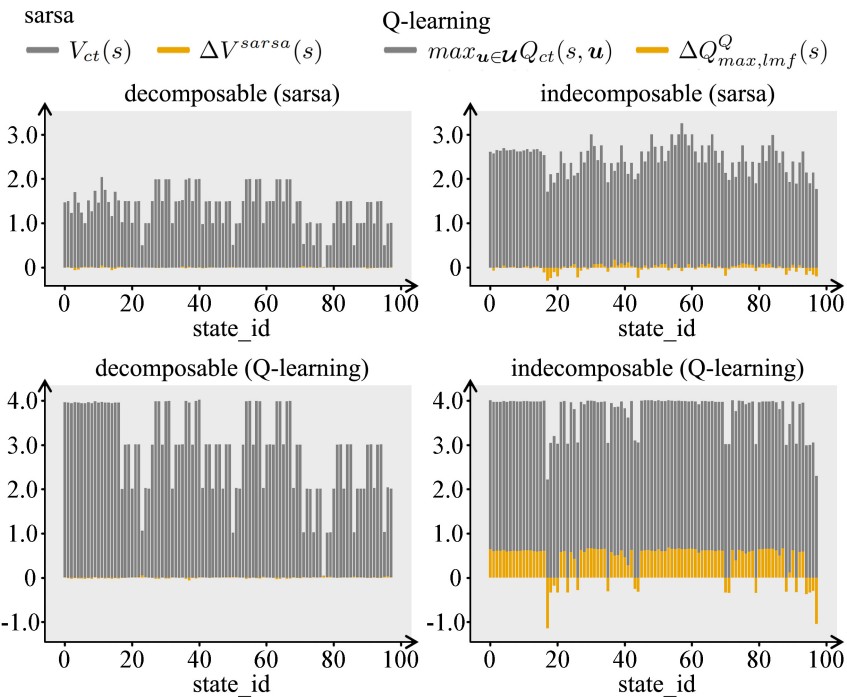

Figure 12: The estimation error of state values (sarsa) and maximal action-values (Q-learning) in decomposable & indecomposable MMDPs, where $\Delta V^{sarsa}(s) = \widetilde{V}_{lmf}(s) - V_{ct}(s)$.

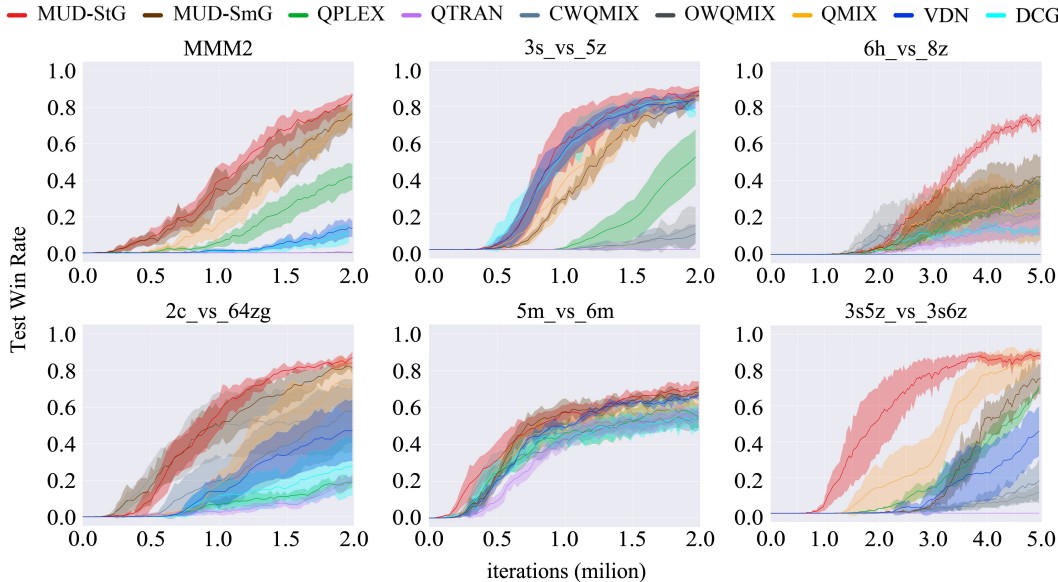

Figure 13: Mean test win rate of value decomposition methods on SMAC.

## K.3   ABLATION STUDIES

We carry out ablation studies to evaluate the effect of gradient shaping. Specifically, we compare two MUD mixing functions with gradient shaping (i.e., MUD-SmG and MUD-StG) and MUD without gradient shaping in predator prey environments. The expression of the mixing function without gradient shaping is $Q_{tot}(s, \boldsymbol{u}) = \sum_{a=1}^{n} Q_a(\tau_a, \boldsymbol{u}_a) - b(s, \boldsymbol{u})$. The experimental results are shown

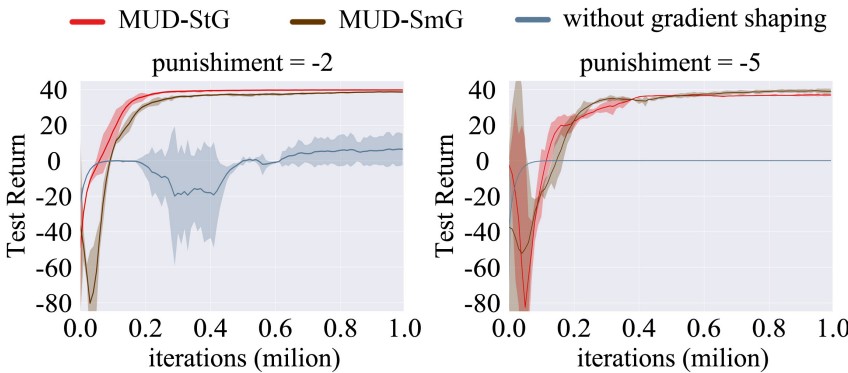

Figure 14: Evaluation of gradient shaping in predator prey environments.

in Fig.14. Besides, we further evaluate the influence of the hyper-parameter $\alpha$ to MUD-StG in predator prey environment with the punishment of -2. The experimental result is shown in Fig.15.

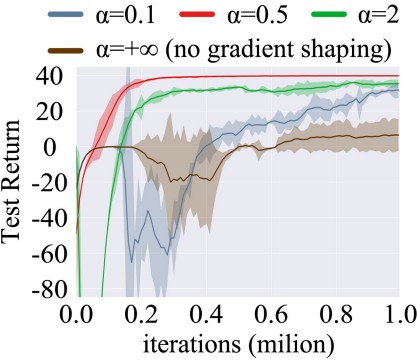

Figure 15: MUD-SmG with different $\alpha$ in predator prey environment with the punishment of -2.

*Difference between decomposable MMDP and decomposable games proposed by previous works.* The decomposability of multi-agent games has also been discussed by previous works. (Dou et al., 2022) defines decomposable game with an unexplainable factorization of transition function. (Castellini et al., 2021) empirically investigates the learning power of different MARL methods on one-shot games. Such works decompose the game agent by agent. In our definition of decomposable MMDP, the game is decomposable by agent groups. Besides, our defined decomposability of MMDP is intuitively explicable. By decomposing the state-action space, the task is decomposed into multiple independent sub-tasks. As a result, the action-value function of the whole task equals the sum of the action-value functions of all sub-tasks. Most importantly, we are the first to prove a sufficient and necessary condition between the decomposability of an MMDP and the linear factorizability of the action-value function.

