# OpenReview forum: "Towards Complete Expressiveness Capacity of Mixed Multi-Agent Q Value Function"
_ICLR.cc/2024/Conference — Submitted to ICLR 2024_

### Official Review · Reviewer_dx4A · 2023-10-23

**Soundness:** 3 good
**Presentation:** 3 good
**Contribution:** 3 good
**Rating:** 5
**Confidence:** 4

**Summary:**

This paper aims to tackle the issues of value decomposition. Initially, it explores the expressive capacity of linear mixing functions and then develops a decomposition structure to achieve complete representational capabilities. Additionally, it identifies and offers a solution to the problem of optimal representation interference. Experimental results substantiate the efficacy of the proposed method.

**Strengths:**

The paper presents a substantial and meticulously detailed theoretical derivation and proof.
The proposed method to construct the complete representational structures is interesting and novel.

**Weaknesses:**

The paper suffers from some unclear expressions and inconsistencies with prior research.
The experimental evaluation is lacking.
See questions below for detail.

**Questions:**

1. The relationship between this paper and [1] requires further elucidation. What's the connection between "the LMF in indecomposable MMDP has unbaised TD target in sarsa" and "the on-policy LMF with fitted-q-iteration has at least one fixed-point Q-value that derives the optimal policy" ? It appears that the results in [1] might be stronger.

2. The paper mentions that addressing single-step matrix games is applicable to solve the optimal policy, but only for sarsa. Does this imply that the theoritical results can only be upheld through on-policy training, or is this just a conclusion unrelated to the subsequent paper?

3. Why is Eq6 used to define complete representational capacity in this paper? The meaning of complete representational capacity under IGM remains unclear. Is this consistent with prior works?
QTRAN is known be necessary and sufficient for IGM, but it has incomplete expressiveness in the context of this paper. Does this imply that methods with incomplete expressiveness can also theoretically guarantee optimal outcomes?

4. Confusing about the "monotonic mixing" and "strictly monotonic mixing". I think the original "monotonic" in previous work means "strictly monotonic" in this work, rendering the "monotonic" in this work redundant. To the best of the reviewer's knowledge, none of the existing methods under IGM are non-monotonic. In fact, non-monotonic mixing might not satisfy IGM.

5. The paper introduces multiple mixers to cover a wider range of mixing functions. This needs further clarification. Is this theoretically necessary, or is it merely a technique for improved performance? It would be valuable if an ablation study on different numbers of mixers in SMAC is conducted.

6. The meaning of Eq11 requires further explanation. Why is the optimal representation ratio defined as such, and what does a low w* signify in terms of ORI?

7. The final algorithm, particularly the expression of Q_tot, is unclear. How are Eq12 and Eq13 applied in Eq10?

8. Does the final algorithm possess complete expressiveness or is it necessary and sufficient for IGM?

9. Why is the single-step matrix game trained in an on-policy manner? Can the proposed method converge to the optimum through off-policy training? Additionally, can QTRAN achieve convergence to the optimum in this game?

[1] Wang et al. Towards understanding linear value decomposition in cooperative multi-agent q-learning. 2020.

---

> ### Author Response · Authors · 2023-11-22
> **Responses to Reviewer dx4A (Part 1)**
>
> **1. Relationship between this paper and a previous work**
>
> This paper and a previous work [1] investigate the representational limitation of LMF from different perspectives and reach different conclusions.
>
> [1] focuses on the joint Q value function of LMF, exploring the properties of LMF under representational limitation.
> Our paper focuses on the action-value function of MARL tasks, exploring the task property to predict representational limitation.
>
> The conclusion "the on-policy LMF with fitted-q-iteration has at least one fixed-point Q-value that derives the optimal policy" refers to that multi-agent Q-learning with LMF has a convergent region, in which all included value function induces optimal actions.
>
> The conclusion "the LMF in indecomposable MMDP has unbaised TD target in sarsa" refers to that the representational limitation does not change the TD target of sarsa value iteration, which indicates the effect of representational limitation is limited within current time-step. In this case, the task can be decomposed by time steps into multiple one-step games.
>
> The differences between two conclusions lie in:
> 1. the former conclusion is about Q-learning while the latter conclusion is about sarsa;
> 2. the former conclusion describes the convergent property of LMF while the latter describes a possibility to decompose the task by time steps.
>
> We would add the discussion of the differences between [1] and our work to the related works.
>
> **2. Conclusion of "addressing single step matrix games is enough to solve the optimal policy for sarsa"**
>
> This is an conclusion unrelated to the subsequent paper.
>
> **3.1 Eq.6 as the definition of complete representation capability**
>
> As we defined in Eq.5, we use the value range of function to represent its "representational capacity". A function with complete representational capacity should be able to represent any value from $-\infty$ to $+\infty$. Notice the optimal action-value $\mathcal{Q}(s,u^*)$ is the largest. Therefore, for a mixing function with complete representational capacity, we require $\forall s,u\in\mathcal{S}\times\mathcal{U}$, the value range $R(f)=(-\infty, \mathcal{Q}(s,u^*)]$.
>
> More details about Eq.6:
>
> Note that Eq.6 can *not* be reached by SMMFs. Detailed proof is available in Section 4.1. Here we provide an intuitive example:
> 1. Problem setting: agent number = 2, local action space = $\{1,2\}$, one state (omitted)
> 2. Assume: greedy action = $(1,1)$. We have $Q_1(1)>Q_1(2)$ and $Q_2(1)>Q_2(2)$
> 3. According to the definition of SMMF (Eq.4), we have $Q_{tot}(2,2)\leq Q_{tot}(2,1)$ and $Q_{tot}(2,2)\leq Q_{tot}(1,2)$
> 4. which equals: $R(Q_{tot}(2,2))=(-\infty,min{Q_{tot}(1,2),Q_{tot}(2,1)})\subsetneqq(-\infty, \mathcal{Q}(s,u))$
> 5. which violates Eq.6.
>
> **3.2 Discussions to QTRAN: optimal outcomes without complete expressiveness**
>
> Although QTRAN is able to learn the optimal policy without complete expressiveness theoretically, QTRAN is still less effective than the methods with complete expressiveness.
>
> The mixing function defines the mapping from local Q value functions $\{Q_1(\tau_1,u_1),\cdots,Q_n(\tau_n,u_n)\}$ to the joint Q value function $Q_{tot}(\tau,u)$.
> In QTRAN, we have $Q_{tot}(\tau,u)=\sum_{a=1}^n Q_a(\tau_a,u_a)$. Therefore, QTRAN is an application of LMF.
>
> As we discussed in related works (the section paragraph, Appendix G.2), QTRAN applies the LMF in a special way. QTRAN introduce two auxiliary networks to train the joint Q value function. Firstly, QTRAN introduce a central Q value function $Q_{ct}(\tau,u)$ (with complete expressiveness) to approximate the action-value function.
> Secondly, QTRAN introduce a ventral state value function $V_{ct}(\tau)$ to represent the difference between $Q_{ct}(\tau,u_{gre})$ and $Q_{tot}(\tau,u_{gre})$. $Q_{tot}(\tau,u)$ is trained with the target $y:=Q_{ct}(\tau,u)-V_{ct}(\tau)$ and would not updated if $Q_{tot}(\tau,u)>y$.
>
> To sum up, QTRAN does not apply a mixing function with complete expressiveness. As a result, to overcome the non-optimality caused by the representational limitation, QTRAN requires auxiliary networks. These networks increase the error of value representation. Specifically, QTRAN need to train the auxiliary networks $Q_{ct}(\tau,u)$ and $V_{ct}(\tau)$ at first, and then train the joint Q value function with the target $y:=Q_{ct}(\tau,u)-V_{ct}(\tau)$. The training error of $Q_{ct}(\tau,u)$ and $V_{ct}(\tau)$ would accumulate to the train of $Q_{tot}(\tau,u)$. By contrast, the joint Q value function with complete expressiveness can be trained end-to-end directly by value iteration, which is more efficient.
>
> We would add the discussions above to the related works.
>
> *Reference:*
>
> [1] Wang et al. Towards understanding linear value decomposition in cooperative multi-agent q-learning. 2020.

---

> ### Author Response · Authors · 2023-11-22
> **Responses to Reviewer dx4A (Part 2)**
>
> **4.1 Confusing about "monotonic" and "strictly monotonic"**
>
> There is only the concept of "monotonic" in previous works, which is introduced by QMIX. Let denote it by "original monotonic" (OM for brevity).
> The definition of OM dose not match the reference of it, for which we separate the concept into two part, i.e., "monotonic" (M for brevity) and "strictly monotonic" (SM for brevity).
>
> The OM in previous works equals M in definition but actually refers to SM.
> In most cases, the OM in previous works refers to the QMIX mixing function.
> In QMIX, the OM is defined as $\frac{\partial Q_{tot}(s, \tau, u)}{\partial Q_a(\tau_a,u_a)}\geq 0$, which is the same as the definition of M (Eq.3) in our paper. But according to our definition of SM (Eq.4), the mixing function of QMIX (which equals to the SMMF, Table 2) is actually SM.
>
> We would supplement the intention of separating the OM into M and SM in Section 2.2.
>
> **4.2 All OM satisfies the IGM**
>
> As we discussed in Section 2.2, "strictly monotonic mixing functions naturally satisfy the IGM (the proof is available in Appendix A)".
> Since the OM in previous works refer actually to the SM, as the reviewer mentioned, in the view of previous works, all OM functions satisfy the IGM.
>
> **4.3 redundant definition of "monotonic"**
>
> We need to find M but non-SM functions because:
> 1.  M functions have better convergence property than non-M (as shown in Fig.10).
> 2.  SM functions suffer from the representational limitation due to the bounded difference.
>
> Such functions can not be described with out the definition of M.
>
> **5. Intention to introducing multiple mixers**
>
> We introduce multiple mixers for two reasons:
> 1. Multi-channel mixing (i.e., mixing by multiple mixers) enable the MUD structure covering a broader range of mixing functions. For example, QPLEX applies n-channel mixing.
> 2. Multi-channel mixing sometime has better performance than single-channel mixing. We would supplement the results of empirical studies on this problem in the Appendix.
>
> **6. Further explanation of Eq.11 (optimal representation ratio)**
>
> We define the optimal representational ratio as the relative representational weight of the optimal action-value.
>
> According to the illustration of representational cross interference in Fig.4, $Q_{tot}(s,1,1)=F \left(Q_1(s,1), Q_2(s,1)\right)$ and $Q_{tot}(s,1,2) =  \left(Q_1(s,1), Q_2(s,2) \right)$ shapes each other through the gradient on the common term $Q_1(s,1)$. Therefore, the gradient $F_a '(s,1,1) = \frac{ Q_{tot}(s,1,1)} {Q_1(s,1)} $ can be viewed as the representational weight of $Q_{tot}(s,1,1)$.
> Besides, the representational weight of an action-value also depends on its probability $\pi(u|s)$.
>
> Note that $Q_{tot}(s,u^*)=F (Q_1(s,u_1^*), \cdots, Q_n(s,u_n^*))$.
> In Eq.11, $w_a^*$ denotes the relative representational weight of the optimal action-value $Q_{tot}(s,u^*)$ on the term $Q_a(s,u_a)$, where
> 1. the expectation of gradient $\pi(u^*|s) \cdot F_a '(s,u^*)$ denotes the representational weight of the optimal Q value $Q_{tot}(s,u^*)$;
> 2. the denominator of $w_a^*$ is applied for normalization;
> We calculate the average relative representational weight across all terms of $Q_{tot}(s,u^*)$ by $u^*=\frac{1}{n}\sum_{a=1}^n w^*_a$.
>
> A larger optimal representation ratio indicates a larger relative representational weight of the optimal action-value. In this case, the representation of optimal action-value would be less affected by the others. We will supplement more details of $w^*_a$ in our paper.

---

> ### Author Response · Authors · 2023-11-22
> **Responses to Reviewer dx4A (Part 3)**
>
> **7. Unclear final algorithm**
>
> Taking Eq.12 as an example, we have $Q_{tot}(s,u)=Q_{tot}(s,u_{gre})-\Delta F=Q_{tot}(s,u_{gre})-e^{-b_\psi(s,u)}\cdot \frac{1}{n} \sum_{a=1}^n \left(1-\frac{Q_a}{Q_{a,gre}}\right)-b_\psi(s,u)$
>
> where $ b_\psi(s,u)=|\hat b_\psi(s,u)- \hat b_\psi(s,u_{gre})| $. $\hat b(s,u)$ can be directly implemented by an MLP. Notice $Q_{tot}(s,u_{gre})$ is equivalent to $V(s)$. Therefore, we have $Q_{tot}(s,u)=V(s)-e^{-|\hat b_\psi(s,u)- \hat b_\psi(s,u_{gre})|}\cdot \frac{1}{n} \sum_{a=1}^n \left(1-\frac{Q_a}{Q_{a,gre}}\right) -|\hat b_\psi(s,u)- \hat b_\psi(s,u_{gre})|$.
>
> We can compound $d$ strictly monotonic mixing functions as $Q_{tot}(s,u)=V(s)-e^{-|\hat b_\psi(s,u)- \hat b_\psi(s,u_{gre})|}\cdot \frac{1}{d}\sum_{i=1}^d\frac{1}{n} \sum_{a=1}^n \left(1-\frac{Q_{a,i}}{Q_{a,gre,i}}\right)-|\hat b_\psi(s,u)- \hat b_\psi(s,u_{gre})|$
>
> We will supplement the implementation details of our proposed mixing functions in the Appendix.
>
> **8.1 Whether final algorithm possess complete expressiveness**
>
> The final algorithms have complete representational capability because they apply the MUD structure.
>
> **8.2 Whether final algorithm is necessary and sufficient for IGM**
>
> The final algorithms are sufficient but not necessary for IGM. As shwon in Table 1 (Appendix G.2), the final algorithms has many good properties such as monotonicity, satisfying the IGM, complete representational capacity and addressing ORI.
>
> **9.1 On-policy training of matrix game**
>
> The condition of on-policy data distribution is only necessary for Theorem 3.3.
> In words, on-policy training is not necessary for the proposed methods, which are also able to converge to the optimum for off-policy training.
>
> **9.2 Performance of QTRAN**
>
> As we tested, QTRAN is able to tackle this task. According to our response to Question 3.2, QTRAN is an application of LMF. We do not present the evaluation results of QTRAN in this task because we only evaluate the performance of different mixing functions here, where the mixing function of QTRAN (LMF) is evaluated.
>
> We would supplement the evaluation results of QTRAN and discussions in the context.

---

### Official Review · Reviewer_mg9a · 2023-10-31

**Soundness:** 2 fair
**Presentation:** 3 good
**Contribution:** 2 fair
**Rating:** 5
**Confidence:** 3

**Summary:**

This paper focuses on addressing the important problem of representational limitation in value decomposition methods for Multi-Agent Reinforcement Learning (MARL) problems. The contributions of the paper are as follows:

1. The paper defines the circumstances under which the linear representational limitation occurs, specifically for Linear Mixing Function (LMF).

2. It introduces a two-stage mixing framework called Mixing for Unbounded Difference (MUD) that addresses the representational limitation by ensuring complete representational capacity under the Independent Global Max (IGM) constraint.

**Strengths:**

This paper exhibits several notable strengths across multiple dimensions.

_Originality_

   - The introduction of the Mixing for Unbounded Difference (MUD) framework, as far as the reviewer is concerned, represent new solutions to address the representational limitation issue. (Although it shares some similarities with other methods, see the weakness section).

_Quality_

   - The proposed MUD framework is supported by mathematical reasoning, enhancing its quality as a potential solution.

**Weaknesses:**

1. The first contribution about LMF and the second contribution on SMMF seems to be separated.

2. The findings about LMF is not quite surprised, as previous work indicates its limitations. What could be interesting is why LMF can work well (empirically) on many tasks, especially when used with gradient-based RL methods.

3. Why do the experiments demonstrate advantage of the proposed method on SMAC, but the performance is similar to baselines in relatively easy task of Predator-and-prey?

4. (*) The proposed MUD share similarities with the QPLEX framework. So the empirical comparison is very important. The reviewer is curious why the performance of QPLEX on SMAC is __significantly__ different from what was reported in the original paper.

4.1 Please discuss the difference from QPLEX in detail.

5. (*) The representational interference problem is not unique to the MUD framework and has been discussed by previous work [1].

(4 and 5 are the main reasons for the overall negative score.)

[1] Ye, J., Li, C., Wang, J. and Zhang, C., 2022. Towards global optimality in cooperative marl with sequential transformation. arXiv preprint arXiv:2207.11143.

**Questions:**

Please see the weakness section.

---

> ### Author Response · Authors · 2023-11-22
> **Responses to Reviewer mg9a (Part 1)**
>
> **1. Separated contributions**
>
> The two contributions are coherent logically in the flow of the paper.
> Our paper focuses on the issue of the representational limitation of the Strictly Monotonic Mixing Function (SMMF), where the answers of two key questions remain unclear for the community.
> The first question is: is the representational limitation a frequent problem for SMMF? If so, it is necessary to solve it, which result in the second question: how the limitation arises and how to solve it?
>
> Our first contribution answers the first question, by which we reach the conclusion that the representational limitation is a frequent and notable problem for LMF.
> Our second contribution answers the second question. We prove that the representational limitation of SMMF arises from the bounded difference and propose a unbounded mixing framework.
>
> **2.1 Not quite surprising findings about LMF**
>
> Although previous works indicate the limitation, we are the first to give a sufficient and necessary condition to the occurrence of the limitation. The novelties of our findings lie in
> 1.  We provide a prospective that the representational capacity of a value function depends not only on the function expression, but also on the task settings. We also propose an approach to investigate whether the task settings match the value function in reinforcement learning.
> 2. Since we have proved a necessary condition of the limitation, given the task settings, we can directly judge whether the limitation occurs by the definition of the decomposable MMDP.
> 3. We also reach some interesting conclusions such as: the representational limitation does not change the TD target of sarsa value iteration, which indicates the effect of representational limitation is limited within current time-step. In this case, the task can be decomposed by time steps into multiple one-step games.
>
> **2.2 Why LMF work well empirically**
>
> LMF works well empirically on many tasks because of two good properties, which have been discussed in our paper.
> 1. LMF satisfies the IGM, which ensures the decentralized executions (according to the largest local Q values) always bring the best estimated group interests (i.e., the largest joint Q values).
> 2. LMF is monotonic. As shown in Fig.10 (Appendix H), the monotonic mixing function has better convergence property than the non-monotonic ones.
>
> **3. Advantages of proposed methods on SMAC but similar performance on predator-prey**
>
> Because these two benchmarks are different in evaluation orientations. Experiments on both benchmarks demonstrate the advantages of our algorithms in different perspectives.
>
> SMAC is a complex task involving a variety of strategies such as focusing fire, adjusting formation, utilizing terrain and unit features. These rich strategies increase the tiers and gaps of algorithms' evaluation.
>
> By contrast, predator-prey is a task with a single strategy, i.e., capturing at the same time. As a result, the performances of algorithms are two-tiered. The algorithms tackling this strategy achieve the similar performance with largest return while the algorithms failing to tackle the strategy receive the return of 0.
>
> Our algorithms are the only ones successful to tackle the strategy under the punishment of -5, although our algorithms perform similar with some of the others in predator-prey of punishment -2.

---

> ### Author Response · Authors · 2023-11-22
> **Responses to Reviewer mg9a (Part 2)**
>
> **4.1 Performance difference of QPLEX with original paper**
>
> We apply the QPLEX code from https://github.com/oxwhirl/pymarl, which is a wildly used MARL library. The algorithm of QPLEX is trained and tested under the default settings. As reported by the author of the library in the paper of WQMIX [2], QPLEX performs not as well as the original paper.
>
> The poor peformance of QPLEX could arises from:
> 1. We do not have the training seed of the original curve. As we can see from the shadow of the QPLEX training curve in Fig.13, the performance of QPLEX demonstrates a large variance in SMAC.
> 2. Altough QPLEX is a significant work which firstly achieves complete expressiveness under the IGM constraint, it has some problems, as we response to the question 4.2.
>
> **4.2. The difference between proposed MUD with QPLEX**
>
> Both QPLEX and our methods apply the MUD structure. The differences between QPLEX and our methods lie in
>
> 1. QPLEX suffers form low sample efficiency. The representation of the action-value with poor performance has little contribution to the optimal policy, which should be assigned with small weights. However, QPLEX treats all action-values equally, leading to low sample efficiency. By contrast, our methods are more efficient since they put larger weights on the action-values with better performance.
> 2. QPLEX does not address the problem of optimal representational interference. Our proposed MUDs address the problem by reweighting the representation of action-values.
>
> We would add detailed discussions to these differences to the related works.
>
> **5. Representational interference has been proposed by previous works**
>
> The problem proposed and addressed by TAD [1] (the paper mentioned by the reviewer) is different from the Representational Cross Interference (RCI) proposed in our paper.
>
> In a nutshell, TAD reveals a convergence problem of decentralized execution under gradient descent.
> TAD finds that the policy could be trapped in local optimums under a small learning rate when initialized at the following case:
> a better coordination requires updating the local policies of more than one agent, while decentralized updating each agent's local policy could result in a worse coordination.
>
> By contrast, the RCI describes the cross interference of the representation between different action-values. according to Fig.4, $Q_{tot}(s,1,1)=\mathcal{F}\left(Q_1(s,1), Q_2(s,1)\right)$ and $Q_{tot}(s,1,2)=\mathcal{F}\left(Q_1(s,1), Q_2(s,2)\right)$ shapes each other through their gradients on the common term $Q_1(s,1)$.
>
> To sum up, the differences between the Sub-Optimal Convergent (SOC) problem proposed by TAD and RCI proposed by our paper lie in:
> 1. SOC depends mainly on initialization and learning rate, while RCI depends mainly on sample distribution and the structure of mixing function.
> 2. SOC occurs when updating the policies involving more than one agent, while RCI occurs under samples with shared local actions.
>
> We would add the discussions of TAD to the related works.
>
> *References:*
>
> [1] Ye, J., Li, C., Wang, J. and Zhang, C., 2022. Towards global optimality in cooperative marl with sequential transformation. arXiv preprint arXiv:2207.11143.
>
> [2] Tabish Rashid, Gregory Farquhar, Bei Peng, and Shimon Whiteson. Weighted qmix: Expanding monotonic value function factorisation for deep multi-agent reinforcement learning. Advances in neural information processing systems, 33:10199–10210, 2020a.

---

### Official Review · Reviewer_weNa · 2023-11-01

**Soundness:** 3 good
**Presentation:** 4 excellent
**Contribution:** 3 good
**Rating:** 6
**Confidence:** 5

**Summary:**

The paper studied the problem representation limitation in value decomposition method in fully cooperative multi-agent reinforcement learning. The author proposed a novel idea that the representation limitation comes from two different perspectives, task property and mixing function.

The authors theoretically proved that LMF is free from representational limitation only in a rare case of MARL problems. SMMF suffers from representational limitation due to the bounded difference between the outputs of greedy and current actions.

To address these issues, the authors also proposed a new framework of mixing for unbounded difference and test with experiments on several different environments.

**Strengths:**

This paper studied a very basic problem in value-based multi-agent reinforcement learning, and theoretically proved the limitations of both the problem itself and existing methods.

**Weaknesses:**

The writing quality could be further improved. It's a little bit hard to follow the paper currently.

**Questions:**

1. We still have some other combination types that is stronger than IGM but is not perfect. Please discuss about this part.

---

> ### Author Response · Authors · 2023-11-22
> **Responses to reviewer weNa**
>
> **Discussions to other mixing types**
>
> We have discussed recent value decomposition methods with monotonic mixing functions. As concluded in Table 1, all these methods satisfy the IGM.
> The IGM ensures that the decentralized executions (according to the largest local Q values) always bring the best estimated group interests (i.e., the largest joint Q values).
>
> Although IGM has become a general property in most of recent value decomposition methods, there are also works which decompose the joint Q value function in special ways without the IGM constraint.
>
> For example, Q Path Decomposition (QPD) [1] decomposes the joint Q value function with the integrated gradient attribution technique, which directly decomposes the joint Q values along trajectory paths to assign credits for agents. Being different from other value decomposition works with explicit form of mixing functions, QPD does not restrict the representation relation of the joint Q value function and the local Q value functions. Therefore, QPD does not satisfy the IGM. QPD provide a unique perspective for value decomposition, which learns the mixing function instead of manual design.
>
> Model-Based Value Decomposition (MBVD) [2] propose an implicit model-based MARL methods based on value decomposition. By MBVD, agents can interact with the learned virtual environment and evaluate the current state value according to imagined future states in the latent space. The imagined future states is predicted from the joint historical trajectories, which include all agents' current actions. As a result, the local Q value function depends on the joint action, which could violate the IGM. MBVD introduces environment model in value decomposition, which empower agents with the capability of foresight.
>
> We would add these discussions to the  related works of our paper.
>
> *References*
>
> [1] Yang, Y., Hao, J., Chen, G., Tang, H., Chen, Y., Hu, Y., Fan, C., and Wei, Z. Q-value path decomposition for deep multiagent reinforcement learning. In International Conference on Machine Learning, pp. 10706–10715. PMLR, 2020a.
>
> [2] Xu, Z., Zhang, B., Zhan, Y., Baiia, Y., & Fan, G. (2022). Mingling Foresight with Imagination: Model-Based Cooperative Multi-Agent Reinforcement Learning. Advances in Neural Information Processing Systems, 35, 11327-11340.

---

### Meta-Review · Area_Chair_GSf8 · 2023-12-15

**Metareview:**

This paper considers the value decomposition approach for fully cooperative MARL, and in particular focuses Strictly Monotonic Mixing Function (SMMF). The paper first theoretically shows that Linear Mixing Function (LMF) --- a special case of SMMF, has limited expressiveness ability. Then, the paper proposes a new two-stage mixing framework to improve the expressiveness of SMMF and empirially demonstrate the effectiveness of the proposed methods. After reading the paper, reviews and author responses, I believe several major issues remain to be inherent to this paper, thus recommend rejection: 1. the theoretical result is rather disconnected to the second part of the algorithm. The theoretical result only shows that LMF (a special case of SMMF) has limited representation power, this does not imply SMMF also has limited power, which does not justify the need to improve the representation power of SMMF which is the second part. 2. As reviewers pointed out, there is  significant inconsistency between the performance of an existing algorithm showed in this paper versus the performance reported in prior paper, especially regarding QPLEX. Instead of speculating the reason due to random seed and high variance, the author is encourage to reach out to the authors of QPLEX to investigate the reason for difference and make sure there is no error in the implementation/hyperparameter choice. 3. The writing of this paper should be significantly improved.

**Justification For Why Not Higher Score:**

Theory and empirical results are disconnected. Empirical result is inconsistent with prior papers. Writing is poor.

**Justification For Why Not Lower Score:**

N/A

---

### Decision · Program_Chairs · 2024-01-16

Reject